# RelCon: Relative Contrastive Learning for a Motion Foundation Model for Wearable Data

**Maxwell A. Xu**[1,2*]**, Jaya Narain**[1]**, Gregory Darnell**[1]**, Haraldur Hallgrimsson**[1]**, Hyewon Jeong**[1,3*]**,
Darren Forde**[1]**, Richard Fineman**[1]**, Karthik J. Raghuram**[1]**, James M. Rehg**[2]**, Shirley Ren**[1]

[1]Apple Inc. [2]UIUC, [3]MIT
`maxu@illinois.edu, jnarain@apple.com`

## Abstract

We present RelCon, a novel self-supervised *Rel*ative *Con*trastive learning approach for training a motion foundation model from wearable accelerometry sensors[†]. First, a learnable distance measure is trained to capture motif similarity and domain-specific semantic information such as rotation invariance. Then, the learned distance provides a measurement of semantic similarity between a pair of accelerometry time-series, which we use to train our foundation model to model relative relationships across time and across subjects. The foundation model is trained on 1 billion segments from 87,376 participants, and achieves state-of-the-art performance across multiple downstream tasks, including human activity recognition and gait metric regression. To our knowledge, we are the first to show the generalizability of a foundation model with motion data from wearables across distinct evaluation tasks.

## 1 Introduction

Advances in self-supervised learning (SSL) combined with the availability of large-scale datasets have resulted in a proliferation of foundation models (FMs) in computer vision (Oquab et al., 2023), language (OpenAI et al., 2023), and speech understanding (Yang et al., 2024). FMs provide powerful, general-purpose representations for a particular domain of data, and support generalization to a broad set of downstream tasks without fine-tuning. In contrast, health times-series have not yet benefited from the foundation model approach, with a few exceptions (Abbaspourazad et al., 2024; Pillai et al., 2024; McKeen et al., 2024). This is particularly unfortunate for problems in mobile health (mHealth) signal analysis, which encompasses data modalities such as IMU, PPG, and ECG, as the collection of mHealth data is both time-consuming and expensive (Rehg et al., 2017). However, recent advances in self-supervised learning for health time-series (Liu et al., 2023) have shown promise, raising the question of whether it is now feasible to train mHealth foundation models.

In this paper, we demonstrate, for the first time, the feasibility of developing a foundation model for the analysis of accelerometry data across multiple tasks. Accelerometry is an important mHealth sensor used in human activity recognition (HAR) (Fu et al., 2020), physical health status assessment (Xu et al., 2022), energy expenditure estimation (Stutz et al., 2024), and gait assessment (Apple, 2021), among many other tasks. We use a novel method for self-supervised learning combined with pre-training on a large-scale, longitudinal accelerometry dataset, the Apple Heart and Movement Study (MacRae, 2021). We show that this approach yields an effective representation for accelerometry, useful for multiple downstream tasks, including HAR and gait metric estimation.

The key to our approach is a novel method for self-supervised contrastive learning, RelCon. Our approach exploits the observation that many time-series of interest are typically composed of motifs, short distinctive temporal shapes that may approximately repeat themselves over time (Mueen et al., 2009). An example is the repeating upward swing of the arm while walking, captured by a wrist-worn accelerometer. We learn a motif-based distance function and use this distance to identify positive and negative pairs sampled from within- and between-person time series for contrastive

---

[*]Work done while at Apple

[†]Code implementation can be found at `https://github.com/maxxu05/relcon`

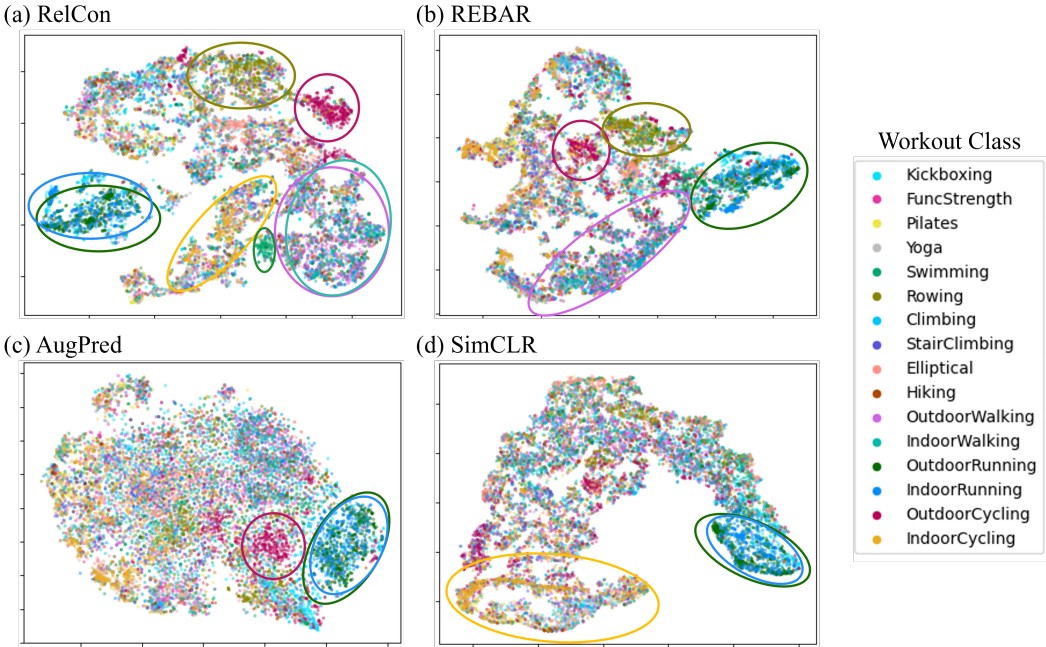

Figure 1: **t-SNE of representation spaces with perplexity=100.** Our RelCon approach has the clearest clusterings based on semantic classes, even forming a specific swimming cluster that is not seen in the other methods. RelCon can also better clearly separate between In/Outdoor Cycling.

learning (Xu et al., 2024). The resulting embedding space captures the natural temporal evolution in physiological states within-person (i.e. fatigue over time) (Glass & Kaplan, 1993), and unique differences in specific physiologies between-persons (Kuncan et al., 2022; Pfeifer & Lohweg, 2018).

In this work, we introduce a novel relative contrastive loss function that models the degree of "positiveness" between the anchor and each candidate, preserving the relative relationships between instances. This helps capture diverse degrees of semantic similarity, reflecting the natural relationships found in the real-world where motions are often similar but never identical. For example, given an anchor signal of "walking," another "walking" signal should be closest, followed by "running," then "cycling," and "yoga" as the most distant. Additionally, we introduce accelerometry-specific augmentations so that our distance measure can learn to compare motifs, while being invariant to the sensor's position and orientation. Our results show that RelCon can capture subtle semantic differences between similar classes (i.e. indoor vs. outdoor cycling), as well as capture the temporally-precise information necessary for gait regression tasks. This is illustrated in Fig. 1, which shows that semantically related classes cluster more effectively in the RelCon feature representation.

The public code repository with the RelCon training set-up, architecture, and evaluation can be found here: `https://github.com/maxxu05/relcon`. Our key contributions are:

1. We introduce RelCon, a novel contrastive learning methodology that models relative differences between instances via a learnable motif-based distance measure. This aims to model the varying degrees of semantic similarity found in real-world motion data.
2. We utilize RelCon to pre-train a foundation model on 1b accelerometry segments across 87k participants from the Apple Heart and Movement Study. This large-scale, longitudinal dataset, sourced from real-world wearable data, enables modeling of human motion dynamics across diverse contexts, supporting robust applications in health and activity monitoring.
3. Our RelCon-trained motion foundation model achieves state-of-the-art performance against 11 models across 6 unique downstream datasets with 4 different tasks, ranging from classification of 16 in-the-wild activities to regression of temporally-precise gait metrics. We are the first to show the generalizability of a motion foundation model across distinct, diverse evaluations.

## 2 RELATED WORK

**Motion time-series FM:** We believe we are the first to train a foundation model for accelerometry data. We define foundation models to be representations that are pre-trained on broad-scale data and are capable of solving multiple diverse downstream tasks without fine-tuning (Bommasani

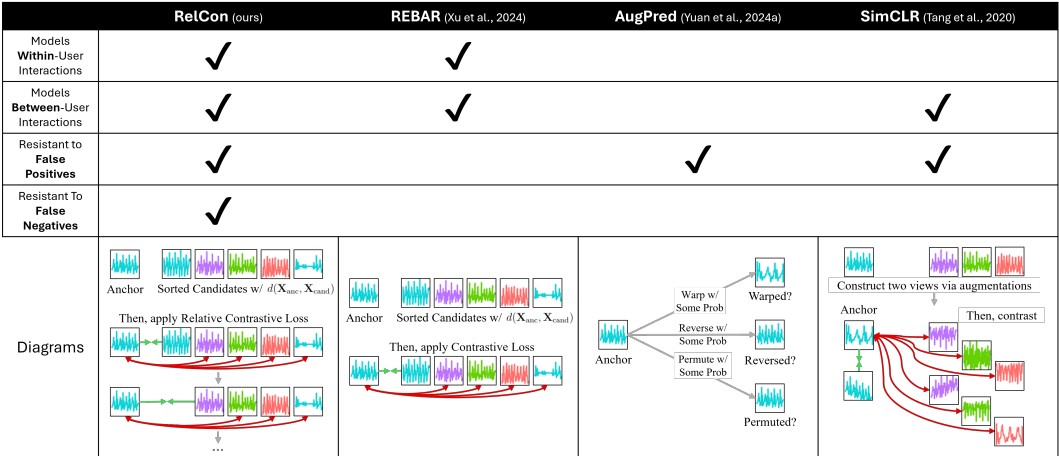

Figure 2: **SOTA Accelerometry SSL Methods.** Each sequence color represents a different user's time-series. RelCon draws candidates from both within- and between-user and ranks them by their relative similarity via a learnable distance function. Then, it iteratively applies a contrastive loss, selecting one candidate as positive while assigning the more distant as negative. This helps prevent false positives/negatives because the full relative ranking is captured. Prior approaches define a single positive/negative set, risking semantic errors if positive/negatives are misdefined. AugPred and SimCLR are resistant to false positives because they construct positive pairs via semantics-preserving augmentations, but REBAR does not have a semantically-constrained pair construction.

et al., 2021) (e.g., each task uses frozen weights with supervised training of a light-weight prediction head). The closest related work is Yuan et al. (2024), which uses 700k person-days of data from UK Biobank to train an accelerometry representation. While their model is evaluated on multiple datasets, all of the tasks are HAR, making it unclear whether the learned representation captures broader motion information. In contrast, we test our FM on multiple gait analysis tasks as well as HAR, demonstrating the effectiveness of our representation for both regression and classification tasks. Narayanswamy et al. (2025) also focuses on HAR but uses minute-level hand-crafted features instead of the raw 100 Hz signals necessary for temporally-precise motion tasks, such as fall detection (Sucerquia et al., 2018) or gait assessment (Alobaidi et al., 2022). Other notable FMs focus on general time-series forecasting (Das et al., 2023; Ansari et al., 2024) or do not model accelerometry motion data (Goswami et al., 2024; Abbaspourazad et al., 2024; Saha et al., 2025; Song et al., 2024).

**Motion time-series SSL:** Our RelCon architecture introduces a novel contrastive learning approach for time-series data. It is most closely-related to REBAR's (Xu et al., 2024) motif-based positive pair generation approach. However, REBAR is not adapted to accelerometry and lacks invariance to well-understood accelerometry-semantic-preserving transformations. Additionally, REBAR employs a "hard" contrastive loss, leading to sensitivity to false positives and negatives due to rigidly designating one positive and all others as negatives. RelCon overcomes these limitations by learning a motif-based distance with accelerometry-specific augmentations and introducing a relative loss function to mitigate false labeling effects. Most prior motion SSL approaches focus exclusively on HAR (Logacjov, 2024; Haresamudram et al., 2024; Straczkiewicz et al., 2021). Haresamudram et al. (2022) benchmarked a number of SSL approaches on accelerometer data, testing generalizability across datasets and sensor positions. The best methods focus on either utilizing augmentations in a SimCLR-fashion (Tang et al., 2020) or utilizing them as prediction targets (Saeed et al., 2019). Comparison of these two models to REBAR can be seen in Fig. 2. Models for other motion-based topics – including walking speed (Shrestha & Won, 2018; Soltani et al., 2021), gait (Slemenšek et al., 2023; Brand et al., 2024), and health monitoring (Takallou et al., 2024) have focused on single-task supervised learning. For alternative biosignals, such as PPG (Pillai et al., 2024), ECG (Jeong et al., 2023; Diamant et al., 2022), and EEG (Pradeepkumar et al., 2025), there have been recent works that develop domain-specific SSL approaches.

**Soft Label Learning Approaches:** A feature of our RelCon approach is the use of soft labels to obtain more fine-grained characterizations of similarity. Most prior self-supervised contrastive methods for time-series have adopted the hard label approach (Yue et al., 2022; Tonekaboni et al., 2021; Zhang et al., 2022), including the previous REBAR method (Xu et al., 2024). One exception is Lee et al. (2023), which proposed a soft contrastive approach for classification and anomaly detection. However, unlike RelCon, they utilize non-learnable distance measures, such as dynamic time warping (Müller, 2007), that cannot be trained to capture domain-specific semantic features.

## 3 METHODOLOGY

The RelCon methodology has two key components. The first is in its innovations to learn a distance measure that captures semantic accelerometry information, described in Section 3.1. The second is a novel relative contrastive loss that encodes relative order relationships, described in Section 3.2.

### 3.1 LEARNABLE DISTANCE MEASURE

Following prior work (Xu et al., 2024), we train a neural network to learn a distance measure to identify whether two sequences have similar temporal motifs (Schäfer & Leser, 2022) and are semantically similar. After training, the architecture is frozen and used as a static function to determine the relative similarities of candidate samples in the RelCon approach. The distance measure architecture is defined in Eq. 1 below:

$$d(\mathbf{X}_{\text{anc}}, \mathbf{X}_{\text{cand}}) := \|\hat{\mathbf{X}}_{\text{anc|cand}} - \mathbf{X}_{\text{anc}}\|_2^2 \tag{1}$$

$$\hat{\mathbf{X}}_{\text{anc|cand}} = = \big((\text{CrossAttn}(\mathbf{X}_{\text{anc}}|\mathbf{X}_{\text{cand}})\mathbf{W}_o + \mathbf{b}_o) + \mu_{\text{cand}}\big)\sigma_{\text{cand}} \tag{2}$$

$$\text{CrossAttn}(\mathbf{x}_{\text{anc}}|\mathbf{X}_{\text{cand}}) = \sum_{\mathbf{x}_{\text{cand}} \in \mathbf{X}_{\text{cand}}} \text{sparsemax}_{\mathbf{x}_{\text{cand}} \in \mathbf{X}_{\text{cand}}}\Big(\text{sim}\big(f_q(\mathbf{x}_{\text{anc}}), f_k(\mathbf{x}_{\text{cand}})\big)\Big) f_v(\mathbf{x}_{\text{cand}}) \tag{3}$$

$$f_{\{q/k/v\}}(\mathbf{X}_{\{\text{anc/cand}\}}) = \text{DilatedConvNet}_{\{q/k/v\}}\left(\frac{\mathbf{X}_{\{\text{anc/cand}\}} - \mu_{\text{cand}}}{\sigma_{\text{cand}}}\right) \tag{4}$$

where $\mathbf{X} \in \mathbb{R}^{T \times D}$ and $\mathbf{x}, \mu, \sigma \in \mathbb{R}^D$ with $T$ as the time length and $D$=3 for our 3-axis accelerometry signals. The distance between an anchor sequence and a candidate sequence, $d(\mathbf{X}_{\text{anc}}, \mathbf{X}_{\text{cand}})$, is defined as the reconstruction accuracy to generate the anchor from the candidate. The distance measure is strictly dependent on the motif similarities between the anchor and candidate that are captured in the dilated convolutions in $f_{\{q/k\}}$ (Xu et al., 2024). See code for further details.

Prior work with this distance measure only captured a simple exact-motif-matching mechanism because the original masked reconstruction training task can be solved by exact-matching the non-masked regions. To enhance the distance measure, we introduce 3 key innovations:
1. Use Accel-specific augmentations (Tang et al., 2020) during training to learn a motif-matching mechanism that is invariant to Accel-semantic-preserving transformations.

During training, we define $\mathbf{X}_{\text{cand}} := Aug(\mathbf{X}_{\text{anc}})$, making the candidate an augmented version of the original sequence, using augmentation procedure from Accel SimCLR (Tang et al., 2020). This prevents an exact-match solution and helps the model learn to reconstruct the original anchor from semantically similar but altered candidates, such as recognizing running signals despite variations like an upside-down wearable or noise from a loose-fitting device.

2. Replace the softmax in the cross-attention with a sparsemax formulation (Martins & Astudillo, 2016) in Eq. 3 to encourage precise motif comparison.

Sparsemax returns the euclidean projection of the unnormalized logits onto the probability simplex, encouraging sparsity (Martins & Astudillo, 2016). This prevents diffuse attention distributions that compare minor, irrelevant motifs within the anchor and candidate. Sparsemax ensures the model reconstructs with distinct features, enabling our measure to capture class-specific information.

3. Use reversible instance normalization (Kim et al., 2021) to normalize an anchor based upon the candidate, preserving relative anchor-candidate magnitude information.

The anchor sequence and final reconstruction output are normalized (in Eq. 4) and unnormalized (in Eq. 2) using candidate sequence statistics. When an anchor and candidate have drastically different magnitudes, it is more likely they represent different fundamental motions and should have increased reconstruction error. Reversible normalization helps preserve this effect.

### 3.2 RELATIVE CONTRASTIVE LOSS: ALL PAIRS ARE POSITIVE ...
BUT SOME PAIRS ARE MORE POSITIVE THAN OTHERS

Normalized temperature cross entropy loss (NT-Xent) is standard in contrastive learning (Eq. 5). It learns a class discriminative embedding space by pulling the anchor and positive instances closer together in the embedding space and pushing all negatives away from the anchor, as in:

$$\ell(\mathbf{X}_{\text{anc}}, \mathbf{X}_{\text{pos}}, \mathcal{S}_{\text{neg}}) = -\log \frac{\exp(\text{sim}(\mathbf{X}_{\text{anc}}, \mathbf{X}_{\text{pos}})/\tau)}{\sum_{\mathbf{X}_{\text{neg}} \in \mathcal{S}_{\text{neg}}} \exp(\text{sim}(\mathbf{X}_{\text{anc}}, \mathbf{X}_{\text{neg}})/\tau) + \exp(\text{sim}(\mathbf{X}_{\text{anc}}, \mathbf{X}_{\text{pos}})/\tau)} \tag{5}$$

where $\text{sim}(\mathbf{X}_{\text{anc}}, \mathbf{X}_{\text{pos}}) = \langle E(\mathbf{X}_{\text{anc}}), E(\mathbf{X}_{\text{pos}}) \rangle$ and $E(\cdot)$ is some encoder function. Traditionally, NT-Xent uses strict, "hard" labels that define absolute positive and negative sets. However, we would like to use the learned distance measure to capture relative ordering among the candidates in the loss. That is, *if $d(\boldsymbol{X}_{anc}, \boldsymbol{X}_i) > d(\boldsymbol{X}_{anc}, \boldsymbol{X}_j)$, the loss learns to preserve that relative ordering in the $E(\cdot)$'s embedding space.* To achieve this, we redefine $\mathcal{S}_{\text{neg}} \coloneqq f_{\text{neg}}(\mathbf{X}_{\text{anc}}, \mathbf{X}_i, \mathcal{S}_{\text{cand}})$ as a function in Eq. 6, allowing the set of negatives to dynamically change based on the current positive pair. This enables the construction of our Relative Contrastive Loss function in Eq. 7:

$$f_{\text{neg}}(\mathbf{X}_{\text{anc}}, \mathbf{X}_{\text{pos}}, \mathcal{S}) = \{\mathbf{X} \in \mathcal{S} : d(\mathbf{X}_{\text{anc}}, \mathbf{X}) > d(\mathbf{X}_{\text{anc}}, \mathbf{X}_{\text{pos}})\} \tag{6}$$

$$\mathcal{L}_{\text{RelCon}} = \sum_{\mathbf{X}_i \in \mathcal{S}_{\text{cand}}} \ell(\mathbf{X}_{\text{anc}}, \ \mathbf{X}_{pos} \coloneqq \mathbf{X}_i, \ \mathcal{S}_{\text{neg}} \coloneqq f_{\text{neg}}(\mathbf{X}_{\text{anc}}, \mathbf{X}_i, \mathcal{S}_{\text{cand}})) \tag{7}$$

This loss iteratively calculates NT-Xent over each candidate, $\mathbf{X}_i \in \mathcal{S}_{\text{cand}}$, iteratively setting each candidate as the positive, $\mathbf{X}_{pos} \coloneqq \mathbf{X}_i$. The negative set for the given positive is then defined as all candidates with a greater distance measure than the given positive $\{\mathbf{X} \in \mathcal{S} : d(\mathbf{X}_{\text{anc}}, \mathbf{X}) > d(\mathbf{X}_{\text{anc}}, \mathbf{X}_{\text{pos}})\}$. Candidates with larger distances from the anchor appear in more negative sets, reinforcing the relative ranking. In this way, the RelCon loss function captures hierarchical structure of similarities in the embedding space based on their relative relationships with the anchor.

In order to capture diverse relationships via RelCon, our pool of candidates originates from two sources: *(1) sampling within the user*, across time and *(2) sampling between users*, within the batch. In *(1)*, we choose $c$ random subsequences from the same user as the anchor sequence. This helps capture within-person temporal dynamics in the encoder, such as how a user's motion signals may indicate fatigue over time. In *(2)*, sampling between-users allows the model to learn physiological similarities and differences across other users. Our RelCon procedure is visualized in Fig. 2.

## 4 EXPERIMENTAL DESIGN

### 4.1 FOUNDATION MODEL PRETRAINING

We trained models on Inertial Movement Unit (IMU) sensor data from the Apple Heart & Movement Study (AHMS) (MacRae, 2021). AHMS is an ongoing research study designed to explore the links between physical activity and cardiovascular health, which is sponsored by Apple and conducted in partnership with the American Heart Association and Brigham and Women's Hospital. To be eligible, participants must — among other eligibility criteria – be at least 18 years of age (at least 19 in Alabama and Nebraska; at least 21 in Puerto Rico), reside in the United States, have access to an Apple Watch, and provide informed consent electronically in the Apple Research app.

The training data included a subset of study data with 87,376 participants recorded over one day, with a 10/3/3 train/val/test split. Following prior convention (Reyes-Ortiz et al., 2015), we use 2.56 seconds of the raw 100 Hz 3-axis x,y,z accelerometry signal of the IMU sensor in the wearable device as input to our embedding model. The model was pre-trained with 8 x A100 GPUs for 24 hours. It iterated over a total of 1 billion unique samples, i.e. $\sim$30k unique participant-days. We used a 1D ResNet-34 encoder backbone (3.9M parameters) with a final global average pooling to generate a 256-dimensional embedding vector.

### 4.2 DOWNSTREAM EVALUATION

We group our downstream evaluations into two sections: Task Diversity and Benchmarking. In total, we compare against 6 different downstream datasets across 4 different types of downstream tasks. We compare our RelCon foundation model against 11 models total: 3 pre-trained from scratch in the Task Diversity Evaluation and 8 from the prior literature in the Benchmarking Evaluation.

In **Task Diversity Evaluation**, we evaluate how well our foundation model's learned embedding generalizes across diverse downstream tasks, in order to understand how SSL via RelCon enables broad applicability. To this end, these downstream tasks include *gait metric regression* for predicting stride velocity and double support time and *human activity recognition* for classifying in-the-wild activities on the subsequence-level and on the workout-level. These tasks are on two separate datasets. Then, we compare our RelCon-trained foundation model against 3 different models, each pre-trained from scratch with different SOTA SSL methodology under the same, controlled conditions.

In **Benchmarking Evaluation**, we compare our foundation model against prior works on standardized, publicly-available human activity recognition benchmarks. Specifically, we compare *against a Large-Scale Pre-trained model (Yuan et al., 2024)* and *against an Accel SSL Benchmarking Study (Haresamudram et al., 2022)* with 7 distinct SSL methods. These datasets evaluate generalizability under data distribution shifts, including differing sensor positions and inference window lengths.

### 4.2.1 TASK DIVERSITY EVALUATION SET-UP

**Baselines:** Following the same exact training conditions as our RelCon foundation model (i.e. AHMS dataset, total training iterations, encoder backbone), we pre-trained 3 other state-of-the-art accelerometry SSL methods from scratch, which can seen below. Visualizations of how each method works can be found in Fig. 2. Further implementation details in Appendix A.2.

- *Accel SimCLR (Tang et al., 2020)* contrasts positive pairs formed by accel-specific augmentations (i.e. 3D rotation) and demonstrates state-of-the-art performance (Haresamudram et al., 2022).
- *Augmentation Prediction (Saeed et al., 2019)* predicts whether or not a specific augmentation was applied. This is a classic accelerometry approach that demonstrates competitive performance (Haresamudram et al., 2022) with good scaling ability (Yuan et al., 2024).
- *Accel REBAR (Xu et al., 2024)* uses a "hard" contrastive loss with learned motif-similarity to identify positive pairs, relying on fixed positive and negative sets without capturing relative positions. For a fairer comparison, we adapt REBAR with the accel-specific enhancements from Section 3.1.

**Gait Metric Regression:** All participants completed proctored overground walking tasks with two mobile devices with IMU sensors at each side of the body in different locations (i.e. at the hip, in a front/back pocket, or in a waist bag) (Apple, 2021). We used the Cohort A participants, which included 359 participants with an average age of 74.7, who provided informed consent. Participants conducted 3 walking tasks: 1) one lap at self-selected speed, 2) four laps at an instructed slow speed, and 3) as many laps as possible within six minutes. Each task was conducted along a straight 12-meter course, with an 8-meter pressure mat placed centrally. Various gait statistics were calculated, such as *double support time* and *stride velocity*, which our models were evaluated on.

Each 2.56s subsequence was matched to the lap aggregated target (e.g. total number of steps or average walking speed). The participants were assigned into 50% train and 50% test randomly based on participant ID, where every lap for a given participant falls into the same split. The training split was used to train linear regression probes on embeddings from the self-supervised models. Metrics were selected following a prior report (Apple, 2021): mean squared error (MSE), std dev of squared error (SDSE), mean absolute error (MAE), std dev of absolute error (SDAE), and Pearson Correlation Coefficient. Mean and standard deviations were calculated by aggregating predictions across all errors of a given user and are related to bias and variance. Correlation is used to assess how each user's average gait metric corresponds to ground truth values. Ranges per metric are obtained by retraining the probe five times and calculating the mean and standard deviation.

**Field Human Activity Recognition:** We evaluated activity classification using self-reported activity labels gathered from in-the-wild, unproctored, field data in AHMS. We used a subset of data with ~2k total users across 16 workouts (full list in Fig. 1). These workouts captured a range of diverse activities (e.g. kickboxing and rowing) that are non-trivial to separate (i.e. outdoor cycling vs. indoor cycling). For evaluation, workouts were class balanced with each including ~22 hours of data, for a total of 310 workout hours. We used a 4/1/5 train/val/test split based on participant ID. We ensured this data did not overlap with the pre-training dataset, preventing any data leakage.

Embeddings are generated for each 2.56s-long subsequence. We evaluate classification on both the *subsequence-level* as well as the *workout-level*. At the workout-level, we predict an activity across each of 2.56s subsequences within the workout, and select the most frequently predicted activity. Including two temporal scales which allow us to evaluate how information learned by the embedding interacts with time aggregation. We report F1, Kappa, Accuracy, and macro AUC metrics following previous work (Haresamudram et al., 2022; Yuan et al., 2024; Xu et al., 2024). Ranges per metric are obtained by retraining the probe five times and calculating the mean and standard deviation.

### 4.2.2 BENCHMARKING EVALUATION SET-UP

**Comparison to a Prior Large-Scale Pre-trained Model:** Yuan et al. (2024) proposes an large-scale model with pre-trained on the UK BioBank (Bycroft et al., 2018) dataset. They are able to

| | Gait Metric Regression (Wrist→Waist) | | | | | | | | | |
| | Stride Velocity | | | | | Double Support Time | | | | |
| Model | ↓ MSE | ↓ SDSE | ↓ MAE | ↓ SDAE | ↑ Corr | ↓ MSE | ↓ SDSE | ↓ MAE | ↓ SDAE | ↑ Corr |
|---|---|---|---|---|---|---|---|---|---|---|
| SimCLR | .0121 ± .0002 | .0234 ± .0019 | .0827 ± .0020 | .0726 ± .0007 | **.8454 ± .0133** | .0016 ± .0001 | .0025 ± .0001 | .0317 ± .0009 | .0254 ± .0006 | .7074 ± .0136 |
| Aug Pred | .0144 ± .0002 | .0241 ± .0003 | .0940 ± .0007 | .0749 ± .0005 | .7950 ± .0035 | .0015 ± .0000 | .0025 ± .0001 | .0296 ± .0002 | .0251 ± .0003 | .7214 ± .0029 |
| REBAR | .0147 ± .0010 | .0285 ± .0035 | **.0818 ± .0042** | .0818 ± .0042 | .7853 ± .0178 | .0016 ± .0001 | .0026 ± .0001 | .0316 ± .0011 | .0254 ± .0008 | .6817 ± .0286 |
| **RelCon (ours)** | **.0115 ± .0005** | **.0190 ± .0013** | .0833 ± .0019 | **.0678 ± .0016** | .8431 ± .0039 | **.0014 ± .0000** | **.0024 ± .0001** | **.0275 ± .0004** | **.0249 ± .0004** | **.7559 ± .0120** |

Table 1: **Gait Metric Regression**. Mean and Standard Deviations of Squared Error and Absolute Error were calculated by aggregating across each user and are related to bias and variance, respectively. The strong consistent performance of RelCon shows that our model is able to effectively capture the user-specific and temporally-precise information needed for gait metric regression.

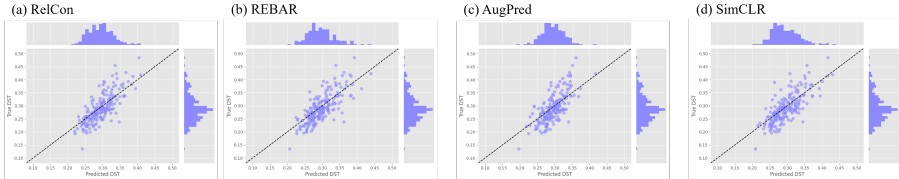

Figure 3: **Correlation between predicted and true DST**. RelCon has the highest correlation.

train over ∼100k unique participants for a total of ∼700k unique participant-days. Their model is based upon an augmentation prediction approach with a ResNet-18 backbone, and they use it to train and embed 10s subsequences of a 3-axis accelerometry signal collected at the wrist.

Following their evaluation methodology, we directly compare our RelCon foundation model against their model along two different dimensions: fine-tuning the entire model and using an MLP probe. We exactly mimic their cross-validation splits on the Opportunity (Roggen et al., 2010) and PAMAP2 (Reiss, 2012) activity classification datasets and use these splits to generate the metric ranges. We also match their 10s length used for inference.

**Comparison to an Accel SSL Benchmarking Study:** Haresamudram et al. (2022) seeks to assess the state of the accelerometry SSL field, in the context of human activity recognition. To this end, they benchmark a diverse range of self-supervised methods, including our aforementioned methods, SimCLR (Tang et al., 2020) and Augmentation Prediction (Saeed et al., 2019), as well as, SimSiam (Chen & He, 2021), BYOL (Grill et al., 2020), MAE (Haresamudram et al., 2020), CPC (Haresamudram et al., 2021), and an Autoencoder (Haresamudram et al., 2019). Each of their approaches uses a unique backbone architecture that corresponds to the original work. They pre-train their SSL methods on the 3-axis wrist-mounted accelerometry signals from Capture-24 dataset (Chan Chang et al., 2021), which is composed of 151 unique participants for a total of ∼4k participant-hours.

Similar to our RelCon method, each of their models are pre-trained on wrist data. However, during evaluation, each activity classification dataset corresponds to accelerometry signals collected at different body positions. We compare our RelCon FM with a downstream dataset at each benchmarked position: HHAR (Blunck et al., 2015) for Wrist, Motionsense for Waist (Malekzadeh et al., 2018), and PAMAP2 (Reiss, 2012) for Leg. In this way, each method will be evaluated on potentially different sensor position as its pre-training data. We adopt the same exact cross-validation splits in the original work and use them to generate our ranges. We match their 2s length used for inference.

## 5 RESULTS AND DISCUSSION

### 5.1 TASK DIVERSITY EVALUATION RESULTS

**Gait Metric Regression:** Table 1 shows the gait metric regression performance of each SSL method.

RelCon had the strongest consistent performance across *both* gait metrics (1st in 3/5 evals for stride velocity and 1st in 5/5 evals for DST). While SimCLR was not specifically designed for gait analysis (Tang et al., 2020), it achieved competitive performance for stride velocity with the highest correlation. This suggests that SimCLR effectively captures overall movement patterns, but may lack the nuanced understanding required for DST, which has more complex gait dynamics compared to velocity. DST measures the brief phase when both feet are on the ground, requiring high temporal resolution to capture subtle transitions between stance and swing, influenced by neuromuscular

| | | Subsequence Level | | | | Workout Level | | |
|---|---|---|---|---|---|---|---|---|
| Model | ↑ F1 | ↑ Kappa | ↑ Acc | ↑ AUC | ↑ F1 | ↑ Kappa | ↑ Acc | ↑ AUC |
| SimCLR | 36.35 ± .42 | **39.32 ± .29** | 37.32 ± .29 | .8372 ± .0004 | 50.06 ± .44 | 49.66 ± .33 | 53.09 ± .30 | .8855 ± .0033 |
| Aug Pred | 34.48 ± .08 | 30.69 ± .12 | 35.02 ± .11 | .8175 ± .0003 | 54.63 ± .79 | 52.44 ± .65 | 55.71 ± .60 | .9049 ± .0010 |
| REBAR | 36.39 ± .28 | 32.75 ± .23 | 36.96 ± .21 | .8334 ± .0011 | 50.29 ± .78 | 48.53 ± .91 | 51.94 ± .87 | .8775 ± .0035 |
| **RelCon (ours)** | **38.56 ± .23** | 35.10 ± .20 | **39.15 ± .19** | **.8417 ± .0009** | **55.28 ± .86** | **53.87 ± .71** | **56.94 ± .68** | **.9078 ± .0016** |

*(Header above table: AHMS Activity Classification (Wrist→Wrist))*

Table 2: **Field Human Activity Recognition across 16 Classes**. RelCon achieves consistently strong performance when evaluated both at the subsequence-level and workout-level. SimCLR does well at the subsequence-level, but performs much worse at the workout-level. The opposite is true for AugPred. This shows that there exist nuances captured by our models at both levels.

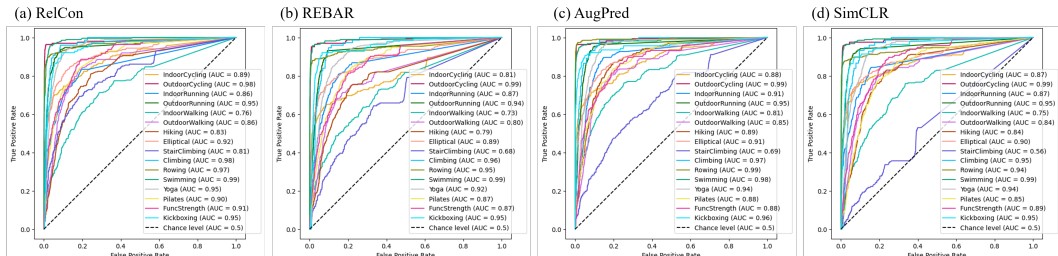

Figure 4: **ROC curves of Workout-Level Field Human Activity Recognition**. Among other gains, RelCon is able to clearly better classify stair climbing (purple) compared to other approaches. Further class-specific prediction results can be found in the confusion matrices found in Fig. 5

coordination (Couto et al., 2023). In contrast, stride velocity is determined by cadence, making it simpler to model via frequency (Kirtley et al., 1985). REBAR, despite using the same learnable distance as RelCon, performed worse across most metrics due to differences in the loss functions (i.e. >.5 difference in corr for both velocity and DST). The hard contrastive loss used in REBAR relies on an absolute set of positives/negatives, limiting its capacity to capture fine-grained relational differences. In contrast, RelCon's relative loss enabled better modeling of subtle differences of transient subject-specific gait dynamics, particularly necessary for modeling the complex neuromuscular functions that control DST, achieving the best performance across all 5 evaluations.

**Field Human Activity Recognition:** Table 2 presents the activity classification performance across subseq and workout levels on real-world field data across 16 classes. Fig. 4 shows per-class performance and Fig. 5 shows detailed class predictions breakdown via a confusion matrix.

Table 2 shows that SimCLR showed strong performance at the subseq-level, with 2nd+ place across the eval metrics. However, performance dropped to 3rd- in workout-level classification. Conversely, AugPred is 2nd place in workout-level, but dropped to the worst method at the subseq-level, across all eval metrics. These differences highlight how varying temporal resolutions capture different aspects of activity dynamics, influencing model performance across time scales.

RelCon continues to achieves state-of-the-art performance with 1st place across 7/8 eval metrics. It also has the best semantic cluster separability in its embedding space, as seen in Fig. 1. As an example, Fig. 4 shows that RelCon is better at classifying stair climbing with a class AUC of .81 vs. .68 / .69 / .56. Fig. 5 shows that the other SSL methods will often confuse stair climbing with climbing or elliptical, but RelCon can capture subtle differences between the slow, deliberate hand swinging motions present in those three classes. Fig. 5 also shows that RelCon more accurately distinguishes between outdoor and indoor running (avg acc normed across the two: .69 vs. .59 / .56 / .68), likely due to its ability to detect the subtle more-uniform motion patterns characteristic of indoor running. Overall, RelCon's strong performance across temporal resolutions and diverse activity categories underscores its robustness and generalizability in activity recognition.

## 5.2 BENCHMARKING EVALUATION RESULTS

**Results from Comparison to a Prior Large-Scale Pre-trained Model:** Table 3 shows that RelCon achieved a stronger performance across all datasets and evaluation methods. We used a ResNet-34 based architecture with a final embedding dimension of 256 and 3.96M parameters, while Yuan

| | | | Opportunity (Wrist→Wrist) | | PAMAP2 (Wrist→Wrist) | |
|---|---|---|---|---|---|---|
| | Eval Method | Pre-train Data | ↑ F1 | ↑ Kappa | ↑ F1 | ↑ Kappa |
| **RelCon FM** | MLP Probe | AHMS | **69.1 ± 8.3** | **62.1 ± 6.2** | **85.4 ± 3.6** | **84.7 ± 3.6** |
| Yuan et al. (2024)'s FM | MLP Probe | UKBioBank | 57.0 ± 7.8 | 43.5 ± 9.2 | 72.5 ± 5.4 | 71.7 ± 5.7 |
| **RelCon FM** | Fine-tuned | AHMS | **98.4 ± 0.9** | **97.9 ± 0.8** | **98.8 ± 1.3** | **98.6 ± 1.6** |
| Yuan et al. (2024)'s FM | Fine-tuned | UKBioBank | 59.5 ± 8.5 | 47.1 ± 10.4 | 78.9 ± 5.4 | 76.9 ± 5.9 |

Table 3: **RelCon FM compared to a Prior Large-Scale Pre-trained Model**. Although the RelCon FM embeds time-series into 256 dim vectors, smaller than Yuan et al. (2024)'s 1024 dim vector, the frozen RelCon FM representation achieves stronger MLP probe performance.

| | | | HHAR (Wrist→Wrist) | Motionsense (Wrist→Waist) | PAMAP2 (Wrist→Leg) |
|---|---|---|---|---|---|
| | | | ↑ F1 | ↑ F1 | ↑ F1 |
| | | **RelCon FM** | **57.63 ± 3.24** | 80.35 ± 0.71 | **53.98 ± 0.76** |
| Self-supervised w/ Frozen Embedding + Linear Probe | Haresamudram et al. (2022) | Aug Pred | 50.95 ± 2.70 | 74.96 ± 1.37 | 46.90 ± 1.14 |
| | | SimCLR | 55.93 ± 1.75 | **83.93 ± 1.78** | 50.75 ± 2.97 |
| | | SimSiam | 45.36 ± 4.98 | 71.91 ± 12.3 | 47.85 ± 2.48 |
| | | BYOL | 40.66 ± 4.08 | 66.44 ± 2.76 | 43.89 ± 3.35 |
| | | MAE | 43.48 ± 2.84 | 61.14 ± 3.45 | 42.32 ± 1.63 |
| | | CPC | 56.24 ± 0.98 | 72.89 ± 2.06 | 45.84 ± 1.39 |
| | | Autoencoder | 53.57 ± 1.14 | 55.12 ± 3.46 | 50.79 ± 1.09 |
| Fully Supervised | | DeepConvLSTM | 54.39 ± 2.28 | 84.56 ± 0.85 | 51.22 ± 1.91 |
| | | Conv classifier | 55.43 ± 1.21 | **89.25 ± 0.50** | **59.76 ± 1.53** |
| | | LSTM classifier | 37.42 ± 5.04 | 86.74 ± 0.29 | 48.61 ± 1.82 |

Table 4: **RelCon FM compared to an Accel SSL Benchmarking Study**. The RelCon FM has the strongest consistent performance, even outperforming the fully-supervised model when the pre-training and target domains match, with both the pre-training dataset and downstream HHAR dataset originating from wrist sensors (Wrist→Wrist).

et al. (2024) used a ResNet-18 based architecture with an embedding dimension of 1024 and 10M parameters. Although our embedding vector had a lower dimensionality, RelCon FM has a stronger performance with the MLP probe (69 vs 57 / 85 vs 73 for Opportunity / PAMAP2). RelCon-learned representations are able to efficiently capture the most important properties of the sequences. Then, with fine-tuning, RelCon can achieve even stronger performance due to the strong initialization.

**Results from Comparison to an Accel SSL Benchmarking Study:** Table 4 compares the RelCon FM against a suite of SSL methods benchmarked in Haresamudram et al. (2022). Our RelCon FM shows strong generalizability, even when there is a sensor position mismatch between training and evaluation, as seen in PAMAP2. When the target dataset's sensor position matches the pre-training domain, as seen in HHAR, RelCon acheives exceptionally strong performance, even beating the fully-supervised methods. Although SimCLR outperforms RelCon in Motionsense, there is a considerable gap between SimCLR/Relcon and the rest of the methods.

## 5.3 ABLATION STUDY RESULTS

**The Importance of Motion Augmentations:** Invariances can capture conceptually important information for motion data, and augmentations have been used extensively in prior approaches to exploit this (Yurtman & Barshan, 2017; Florentino-Liaño et al., 2012; Zhong & Deng, 2014). For example, learning a 3D rotation invariance makes embeddings robust to how users wear their devices. RelCon uses these augmentations while learning the distance function, so that the distance function can identify instances that represent similar characteristics but differ from semantic-preserving transformations. Removing these augmentations from the distance function learning significantly degrades performance (>12% drop in 3/4 tasks, as shown in *w/o Augmentations* in Table 5), because the distance function fails to compare and identify semantically-similar instances.

**Within-Person Dynamics in Time-series:** Variations in time-series signals stem from both within-user and between-user dynamics. Learning embeddings by comparing subsequences within the same user helps capture important temporal dynamics, as shown in the *w/o Sampling Within-Subject* ablation in Table 5. In this ablation, we no longer draw candidates from the within-user, across time. Instead the candidates are drawn from other batch members and augmented versions of them, similar to SimCLR. This leads to a 4% drop in subsequence-level classification (AHMS-Subseq) and a 7% drop in workout-level classification. These results highlight the importance of modeling within-

| | Stride Velocity | Double Support Time | Subseq-level Activity | Workout-level Activity |
|---|---|---|---|---|
| | ↑ Corr | ↑ Corr | ↑ F1 | ↑ F1 |
| **RelCon** | 0.8431 | 0.7559 | 38.56 | 55.28 |
| w/o Augmentations | -5.42% | -13.88% | -12.5% | -17.93% |
| w/o RevIN | -11.66% | -6.01% | -3.81% | -4.11% |
| w/o SparseMax | -0.87% | -3.7% | -5.06% | -7.11% |
| w/o Sampling Within-Subject | -1.40% | -1.40% | -4.49% | -7.25% |
| w/ Softer Metric Loss (Kim et al., 2019) | -3.64% | -6.11% | -1.45% | 1.39% |
| w/ Harder Binary Contrastive Loss (Xu et al., 2024) | -6.86% | -9.82% | -5.63% | -9.03% |

Table 5: **Ablations of Key RelCon Components**. Removing augmentations in our distance learning approach results in a large drop in performance, highlighting the importance of learning a motif-based distance that is robust to accel-specific invariances. RevIN strongly affects the regression tasks. Within-subject interactions and SparseMax are important for classification. Finally, our relative loss strikes a good balance between hardness of the pos/neg psuedo-labels in the loss function.

user dynamics over time, especially for longer-scale workout-level tasks. This is also reflected in SimCLR's performance in Table 2: SimCLR does not capture within-user interactions and also performs worse at the workout-level compare to the subsequence-level task

**Refining our Learned Distance:** Table 5 shows that if we remove the reversible instance normalization, *RevIN*, component or replace the *SparseMax* with the traditional softmax operator within our distance function, then performance suffers. Removal of RevIN causes a 12% drop in the velocity correlation, which shows the importance of capturing relative magnitude for predicting velocity.

**Relative Contrastive Loss is a Good Balance of "Softness":** RelCon's relative contrastive loss is a softer version of REBAR's hard binary contrastive loss. Both use a learnable distance to identify positives and negatives, but REBAR selects only one positive candidate, labeling the rest as negative, increasing the risk of false positives and negatives (Xu et al., 2024). False positives occur when a candidate is incorrectly identified as positive despite belonging to a different class, while false negatives arise when multiple valid positive candidates exist but only one is selected. The softer relative contrastive loss captures the relative "positiveness" of candidates, reducing the impact of false pseudo-labels. As shown in *w/ Harder Binary Contrastive Loss* in Table 5, RelCon's relative contrastive loss improves performance >5% compared to the REBAR binary contrastive loss.

Alternatively, the relative loss function could be softened further with a metric loss function that directly learns the distances in the embedding space to be proportional to distances from the learned motif-based distance function. The distance function is now treated as an exact ground truth. In *w/ Softer Metric Loss* in Table 5, we replace the loss with the log-ratio metric loss, $\ell(\mathbf{X}_{\text{anc}}) = \sum_{\mathbf{X}_{\text{pos}}, \mathbf{X}_{\text{neg}}} \left( \log\left(\frac{\|E(\mathbf{X}_{\text{anc}}) - E(\mathbf{X}_{\text{pos}})\|_2}{\|E(\mathbf{X}_{\text{anc}}) - E(\mathbf{X}_{\text{neg}})\|_2}\right) - \log\left(\frac{d(\mathbf{X}_{\text{anc}}, \mathbf{X}_{\text{pos}})}{d(\mathbf{X}_{\text{anc}}, \mathbf{X}_{\text{neg}})}\right) \right)^2$ from Kim et al. (2019). This results in a performance drop across all tasks, particularly in gait metric regression. This implies that a softer metric loss too exactly matches the distance function, obscuring the subtle features needed to differentiate gait. We aim for a balanced approach that allows the distance function to flexibly capture semantic groups, without strictly constraining the embedding space.

## 6 CONCLUSIONS

We introduce RelCon, the first foundation model for accelerometry data with state-of-the-art performance on multiple diverse downstream motion tasks. RelCon embeddings exhibit consistently strong performance using only simple probes for diverse tasks including stride velocity, double support time, and human activity recognition, on multiple datasets. This demonstrates RelCon's ability to capture fundamental properties of motion that are relevant across tasks. We achieve this through a novel motif-based distance learning approach that incorporates augmentations to learn sensor invariances, along with a relative contrastive loss approach for learning fine-grained interactions between- and within-subjects. Our results show that RelCon is highly effective for motif-based time-series data, such as accelerometry, and could potentially be effective for other quasi-periodic biosignals.

# 7 REPRODUCIBILITY

The code for our RelCon foundation model can be found at `https://github.com/maxxu05/relcon`. This includes the RelCon training methodology, architecture, as well as the reproducible evaluation code from our "Benchmarking Evaluation" task, which utilizes public datasets. This can be used by the broader research community by providing code to easily re-train the RelCon approach in addition to providing a unified benchmarking task to guide model development.

# 8 ACKNOWLEDGMENTS

Thank you to Di Wu, James Ochs, Sunny Chow, and Guillermo Sapiro at Apple for their support in this work. Special thanks as well to Harish Haresamudram for his help in replicating prior results, as well as Catherine Liu for her help in proofreading and editing.

Research reported here was supported by Apple and partially supported by the National Science Foundation Graduate Research Fellowship under Grant No. DGE-2039655 and the National Institutes of Health (NIH) under award P41EB028242. The opinions expressed in this article are the authors' own and do not reflect the views of Apple, NSF, or NIH.

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

# A  Appendix

## A.1  Extra AHMS Field Human Activity Recognition Results

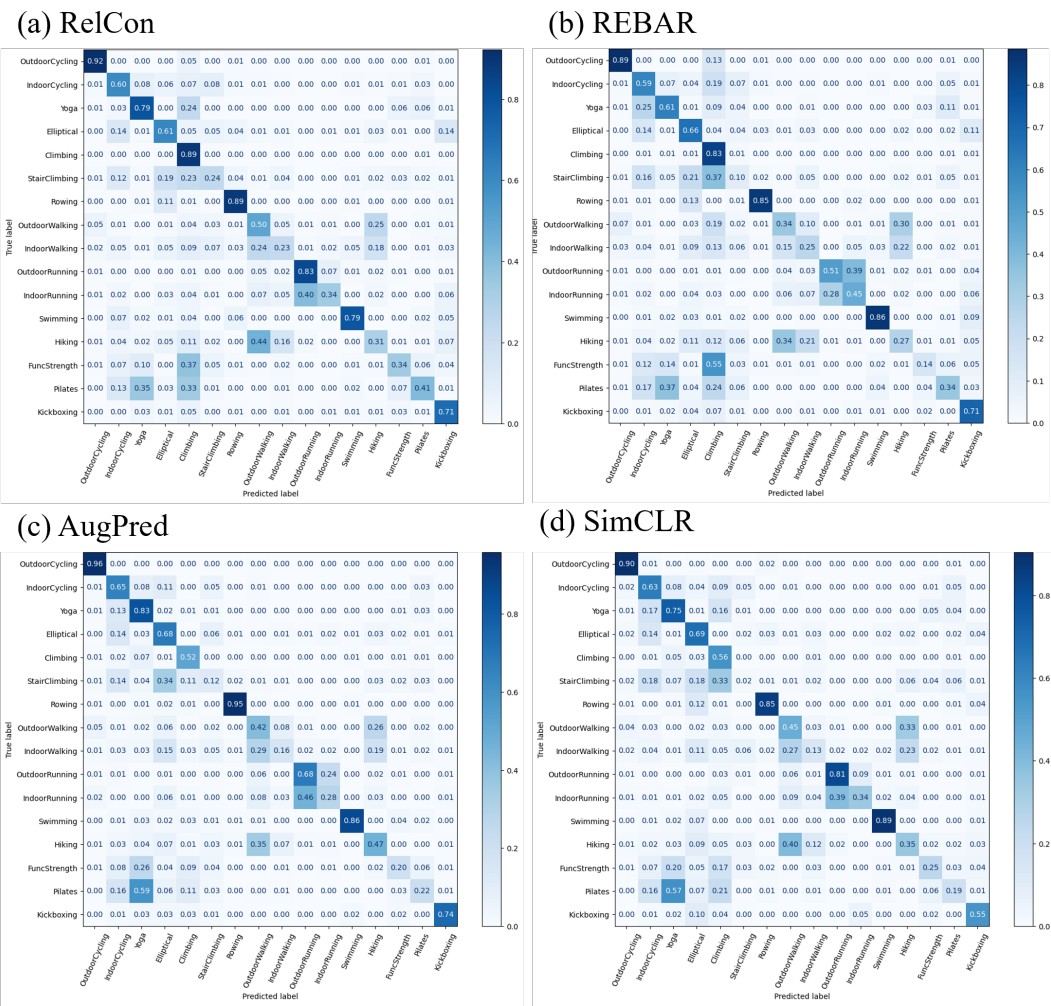

Figure 5: **Confusion Matrices for AHMS Field Human Activity Recognition at the Workout Level.** We can see RelCon has the best performance. Unlike REBAR and AugPred, RelCon can better predict Outdoor running from Indoor Running. RelCon is also able to better predict Stair Climbing unlike the others.

## A.2  Model Implementation Details

**SimCLR**: We follow the approach described by Haresamudram et al. (2023), following the implementation located here: `https://github.com/ubicompsoartutorial/soar_tutorial/tree/main/simclr`. Then we use a batch-size of 64, temperature of 1, and train for 1e5 steps.

**Augmentation Prediction**: We follow the approach described by Yuan et al. (2024), following the implementation located here: `https://github.com/OxWearables/ssl-wearables?tab=readme-ov-file`. Then we use a batch-size of 64 and train for 1e5 steps.

**REBAR**: We follow the approach described by Xu et al. (2024), following the implementation located here: `https://github.com/maxxu05/rebar`. Then we use the prior SimCLR augmentations, a batch-size of 64, temperature of 1, candidate set size of 20, and train for 1e5 steps.

**RelCon**: We use the augmentations utilized by the aforementioned SimCLR model, batch-size of 64, temperature of 1, candidate set size of 20, and train for 1e5 steps. Please refer to our code at `https://github.com/maxxu05/relcon` for further details.

### A.3 DETAILS AND PREPROCESSING ON THE AHMS PRE-TRAINING DATASET

Please refer to original paper on AHMS for further details on the dataset (Shapiro et al., 2023). Specifically, Figure 8 in that work includes visualizations that show general subject demographic distributions including age, body mass index, and self-reported race and ethnicity.

**Preprocessing:** We use the raw 100hz accelerometry data as is, without specific preprocessing techniques, such as filtering or downsampling. We choose to not filter the data as prior work has discouraged filtering in a daily monitoring setting (Campbell et al., 2020), and may prevent our deep learning model from modeling subtle nuances within the signal. Additionally, we would like to develop our methods to be robust to noise via augmentations, such as additive gaussian noise augmentations. We purposely do not attempt to filter out periods of low accelerometry activity. This is because small, minor changes in the accelerometer signal have been shown to still be informative, being able to predict heart rate (Moebus et al., 2024).

### A.4 JUSTIFICATION FOR USING SINGLE-SENSOR-ACCELEROMETER-ONLY DATA

Motion information can be analyzed with multiple sensor streams, whether it be through multiple accelerometer sensors strapped in different locations (Jain et al., 2022) or via extra IMU-based sensors, such as the gyroscopic sensor (García-de Villa et al., 2023). Many prior state-of-the-art human activity recognition, supervised machine learning models will exploit the full sensor suite that includes multi-modality and multi-location sensors (Essa & Abdelmaksoud, 2023; Suh et al., 2023). We recognize the value of a foundation model that can incorporate a multi-modality and/or multi-location stream of data, as it would enable for greater insights on human motion and physiology, and we are interested in investigating this in future work.

However, for our foundation model, we strive for broad generalizability in order to ensure that our learning approach and model is applicable across various settings. This includes low resource settings that only have accelerometer sensors available, as gyroscopic sensors are quite power hungry (Group, 2017). Accelerometer sensors are thus the most common sensor for monitoring human motion (Huang et al., 2023). Additionally, we would like our model to also be applicable for real-world field settings, in which multi-location sensors are uncommon for daily usage due to their bulkiness and discomfort. As such, our foundation model is able to be benchmarked against a broad range of datasets (i.e. our AHMS classification, our Gait Metric Data, HHAR, Motionsense, PAMAP2, Opportunity), which each utilize different sensor hardwares, but all include at least one 3-axis accelerometer sensor.

### A.5 ELABORATING ON THE TABLE IN FIG. 2

False Negatives in a SimCLR-based approach is a well-studied problem, with many recent works proposing novel methodologies to address this (Huynh et al., 2022; Jin et al., 2023; Chien & Chen, 2024). Additionally, the accelerometry SimCLR we are benchmarking (Tang et al., 2020) makes no distinction to model within-user interactions, by treating every subsequence as independent, and hence does not model within-user interactions. Both SimCLR and AugPred creates instances from the original sequence via augmentations, and so they will be resistant to False Positives.

Similar to REBAR, RelCon explicitly models within and between-user interactions by explicitly comparing an anchor against candidates from within-user across time and across other users. However, unlike REBAR, we model the relative positions of our candidates, rather than a binary comparison that treats all negative instances as the same. Then, RelCon is more likely to be resistant to False Positives and False Negatives due to the enhanced comparison that captures the nuanced differences between candidates.

### A.6 Computational Complexity of RelCon

During training, RelCon needs to compare the relative distances of each candidate from the anchor with our distance function. The distance function utilizes a highly parallelizable transformer function with complexity of $O(T^2 \times d)$(Vaswani et al., 2017) and a convolution to embed the inputs with a complexity of $O(k \times T \times d^2)$ (Vaswani et al., 2017). T=256 the subsequence length, d=64 the embedding dimension, and k=15 the kernel size. Therefore, because we calculate this distance function for every candidate, given a size of c, our total complexity during training is $O(c \times T^2 \times d + c \times k \times T \times d^2)$. In our future research, we will work on decreasing the computational cost during training.

### A.7 Evaluation Dataset Descriptions

#### A.7.1 Comparison to a Prior Large-Scale Pre-trained Accel Model (Yuan et al., 2024)

Yuan et al. (2024) has released their code publicly here, which contains exact data split and generations: `https://github.com/OxWearables/ssl-wearables`. Our codebase uses their exact splits[†]. Table 3 in our paper is then constructed by drawing from Table 2 in their original paper. Note that in Yuan et al. (2024), MLP Probe is referred to as "Fine-tuned self-supervised after ConV layers". The used dataset details are as follows:

- **Opportunity** (Roggen et al., 2010): 4-fold leave-one-subject-out cross validation with the sitting, standing, walking, and lying labels.
- **PAMAP2** (Reiss, 2012): 9-fold leave-one-subject-out cross validation with the lying, sitting, standing, walking, ascending stairs, descending stairs, vacuum cleaning, and ironing classes. This is the wrist-specific accelerometry data.

#### A.7.2 Comparison to an Accel SSL Benchmarking Study (Haresamudram et al., 2022)

Although Haresamudram et al. (2022) has not released their code publicly, we contacted the authors directly to ensure that we matched their exact splits and classes evaluated. Our codebase uses their exact splits[†]. Table 4 in our paper is then constructed by drawing from Table 3 and 4 in their original paper. Specifically, HHAR is drawn from Table 4, PAMAP2 from Table 3/4, and MotionSense from Table 3 (Haresamudram et al., 2022). Note that in Haresamudram et al. (2022), Augmentation Prediction is referred to as "Multi-Task Self Supervision". The used dataset details are as follows:

- **HHAR** (Blunck et al., 2015): 5-fold leave-subject-out cross validation with the bike, sit, stairs down, stairs up, stand, and walk classes.
- **Motionsense** (Malekzadeh et al., 2018): 5-fold leave-subject-out cross validation with the downstairs, upstairs, sitting, standing, walking, and jogging labels.
- **PAMAP2** (Reiss, 2012): 5-fold leave-subject-out cross validation with the lying, sitting, standing, walking, running, cycling, nordic walking, ascending stairs, descending stairs, vacuum cleaning, ironing, and rope jumping classes. This is the ankle-specific accelerometry data.

### A.8 Generalizing to Time Lengths Beyond 2.56 seconds

2.56s is a common accelerometry subsequence size for motion tasks (Reyes-Ortiz et al., 2015; Chen & Xue, 2015; Wang et al., 2007; McQuire et al., 2021; Mandong & Munir, 2018). Additionally, we have shown that our approach can easily be used for time-series with other, differing lengths. In our comparisons against Yuan et al. (2024), we utilize a subsequence length of 10 seconds in order to match their evaluations, and we still show strong performance. In our comparisons against Haresamudram et al. (2022), we utilize a subsequence length of 2 seconds in order to match their evaluations. This highlights that our model is robust to varying input lengths and is generalizable across input data configurations, as would be desirable from a foundation model. This flexibility is enabled by the final temporal-global-average-pooling layer that we have at the end of our architecture. Additionally, in our AHMS workout-level classification task, we show how our method can be used with variable-length time-series that can last up to 10 minutes long by aggregating predictions across windows.

