# OpenReview forum: "RelCon: Relative Contrastive Learning for a Motion Foundation Model for Wearable Data"
_ICLR.cc/2025/Conference — ICLR 2025 Poster_

### Official Review · Reviewer_DTZW · 2024-11-03

**Soundness:** 3
**Presentation:** 3
**Contribution:** 3
**Rating:** 6
**Confidence:** 5

**Summary:**

This paper introduces a self-supervised learning approach, RelCon, that uses a relative contrastive learning method to train a motion foundation model for wearable sensor data. The model is pretrained on a large-scale dataset AHMS comprising 1B accelerometer segments from over 87,000 participants. RelCon integrates a learnable distance measure and a softened contrastive loss, which enables the proposed model to capture motif similarity and domain-specific semantic information. The evaluation demonstrates good performance of the proposed method across multiple downstream tasks, including gait metric regression and human activity recognition, compared to existing state-of-the-art approaches.

**Strengths:**

+ This paper presents an extensive dataset for pretraining the model. The dataset scale is substantial, and training on over a billion segments could significantly contribute to wearable motion data analysis.
+ The proposed SSL approach with a learnable distance measure and softened contrastive loss is interesting and looks novel.

**Weaknesses:**

- The authors compared their ReCon foundation model trained on their AHMS dataset using ResNet-34 with Yuan et al. (2024) trained on UkBioBank Dataset using ResNet-18. Even though they argued that ResNet-34 is "smaller and deeper" and performs better than ResNet-18, this is not a fair comparison to show the superiority of their AHMS dataset and method. A fair evaluation should employ the same architecture (e.g., ResNet-18 for both methods) to isolate the effect of the SSL method, rather than introducing variability due to architecture differences.
- The authors do not conduct an adequate benchmarking study on an established dataset for SSL in accelerometer data. For instance, they could evaluate their method by pretraining on the AHMS and Capture-24 datasets, allowing for a more consistent comparison to other SSL methods.

**Questions:**

- Please provide a fair comparison evaluation with Yuan et al. (2024) using the same network architecture.
- Is it possible to evaluate the proposed SSL method with other SSL approaches pretraining on the same dataset like Capture-24?

---

> ### Author Response · Authors · 2024-11-20
> **Response to DTZW (1/1)**
>
> Thank you for your thoughtful and specific feedback, and for recognizing the contribution of a large foundation model for wearables with a novel learning approach.  We appreciate your feedback and have introduced a new experiment that controls for the backbone architecture and continues to demonstrate the superior strength of our RelCon FM compared to the Yuan et al.’s, 2024 FM.
>
> **Re-training our RelCon FM with the same backbone as Yuan et al., 2024 (W1 / Q1)**
>
> - Thank you for the suggestion. We have re-trained our model with the ResNet-18 backbone with a final encoding dimensionality of 1024, and we show the results in the Table below, as well as in Section A.10. Out of the ¾ evaluations , RelCon continues to have stronger performance compared to Yuan et al., 2024’s FM.
> |||||Opportunity (Wrist → Wrist) | | PAMAP2 (Wrist → Wrist) | |
> | :- | :- | :- | :- | :- | :- | :- | :- |
> ||Architecture|Eval Method|Pre-train Data|↑ F1|↑ Kappa|↑ F1|↑ Kappa|
> |RelCon FM|ResNet-18|Fine-tuned|AHMS|**98.1 ± 0.1**|**97.6 ± 1.2**|**97.9 ± 0.5**|**97.5 ± 0.8**|
> |Yuan et al., 2024’s FM|ResNet-18|Fine-tuned|UKBioBank|59\.5 ± 8.5|47\.1 ± 10.|78\.9 ± 5.4|76\.9 ± 5.9|
> |RelCon FM|ResNet-18|MLP Probe|AHMS|**65.4 ± 12.**|**55.7 ± 12.**|62\.5 ± 13.|53\.4 ± 6.6|
> |Yuan et al., 2024’s FM|ResNet-18|MLP Probe|UKBioBank|57\.0 ± 7.8|43\.5 ± 9.2|**72.5 ± 5.4**|**71.7 ± 5.7**|
>
> **Consistent Comparison for Other SSL methods (W2 / Q2)**
>
> - Thank you for this important discussion point. We would like to remark that the “Task Diversity Evaluation” allows us to have a consistent comparison across the RelCon, SimCLR, AugPred, and REBAR SSL methods. Each method is trained with the same total # of training steps, same architecture, and same dataset. SimCLR and AugPred are state-of-the-art SSL methods for IMU-based sensors and REBAR for general time-series signals, so we believe that our experimental set-up is able to sufficiently demonstrate the relative strength of our RelCon method.
> - That being said, we certainly do see the value of re-training our RelCon method on a public, well-established dataset for enhanced reproducibility and easier benchmarking comparisons. We will be releasing the code for our RelCon foundation model publicly upon acceptance. This will include the RelCon training methodology, architecture, as well as the reproducible evaluation code from our “Benchmarking Evaluation” task, which utilizes public datasets. With this, we encourage future work to build on our approach for training foundation models on established SSL datasets.

---

> > ### Comment · Reviewer_DTZW · 2024-11-25
> >
> > I appreciate the authors for addressing my comments.
> > The authors address my comments and mostly answer my concerns.

---

> > > ### Author Response · Authors · 2024-11-25
> > > **Response to Reviewer DTZW**
> > >
> > > Thank you for your response and acknowledgement: it is appreciated!

---

### Official Review · Reviewer_byiS · 2024-11-03

**Soundness:** 3
**Presentation:** 3
**Contribution:** 3
**Rating:** 8
**Confidence:** 4

**Summary:**

This paper presents a relative contrastive learning framework with motion time series data from wearables. Contrastive learning is a common technique for developing pre-trained models for motion time series data. A major component of contrastive learning for motion time series is to construct pair and negative pairs of samples to obtain some representation, which isn’t trivial for motion time series.

The main challenge is the difficulty of generating augmented views of the same time series segment without introducing artifacts. This paper builds upon an existing contrastive framework that makes use of the fact that many short segments of human activities are repeated in nature which can be used as positive pairs e.g. repeated walking, hand-shaking behaviors. The paper improved on a learnable distance metric used to select the positive pairs and introduced a soft contrastive loss so that not all the negative pairs are considered the same.

The experiments presented showed competitive performance against other pre-trained models and other self-supervised methods.

**Strengths:**

1. This paper nicely incorporated characterises of motion time series into the design of the contrastive learning framework which is commendable.
2. The empirical performance is promising.
3. A wide range of benchmarks are included

**Weaknesses:**

1. Neither the code nor the model is open source, preventing future reuse. I believe at least pseudo code should be provided somewhere.
2. The Apple Heart and Movement Study used to retrain the model was collected using consumer-grade devices in a free-living environment. Often, data coming from consumer-grade devices have limited data quality, including noise and missing data. More descriptives on the study participants, data processing pipeline, and an exploratory analysis of the IMU data will help to understand what data goes into pre-training.
3. I believe the results shown in Table 3 when comparing the proposed model and another pre-trained model is biased for several reasons. The benchmark datasets used Opportunity and PAMAP2 are small, it is very easy to overfit ofthese two datasets. The authors are quoting performance in the high 90 of their proposed model which is a sign of overfitting already. Can the authors at least share their code during the review process so that we know how the evaluation was implemented?
4. Another reason the benchmark is overfitted (Table 3) is that the metric difference between a pre-trained model and a non-pre-trained model is a mere 5%. Whereas the performance difference for Yuan et al is about 10%+. Furthermore, Yuan et al, 2024 was pre-trained on the UKB which is 10x the size of the Apple Heart & Movement Study, it is hard to believe the reported performance from this work is about 100% better in some cases.

**Questions:**

1. In text citations have not be been correctly phrased throughout the manuscript
2. The selection of 2.56 second as the window length requires justification. If the proposed work were to be a foundation model, it should be designed to cater for common window lengths used 1s, 5s, 10s etc.
3. Can you share your code at least during the review process as again there are questions around your evaluation procedure which is likely due to overfitting?
4. How are your workout labels obtained, self-reported or via some objective alternative modality?
5. What are the population characteristics of the pre-training population?
6. How did you process your IMU data, filtering, downsampling, non-wear detection?
7. In table 3, can you include a statement to indicate the results from Yuan et al are quoted as you didn't implement their evaluations to avoid confusion.

---

> ### Author Response · Authors · 2024-11-20
> **Response to byiS (1/3)**
>
> Thank you for your careful and thoughtful feedback, and for recognizing the strength of our RelCon method in exploiting time-series properties, as well as our comprehensive experimentation. We can now confirm that we will release our code publicly after publication, and in response to the other feedback, we have revised our manuscript to capture our discussions.
>
> **Code Release for Reproducibility (W1, W3, Q3)**
>
> - Thank you for your advocacy for reproducibility and transparency. We will release the code for our RelCon foundation model publicly upon acceptance. This will include the RelCon training methodology, architecture, as well as the reproducible evaluation code from our “Benchmarking Evaluation” task, which utilizes public datasets. Due to internal constraints, we could not confirm a code release at submission but we can do so now. We believe that this will greatly benefit the broader community by providing code to easily re-train the RelCon approach in addition to providing a unified benchmarking task to guide model development.
>
> **Data Description and Preprocessing (W2, Q4, Q5, Q6)**
>
> - Thank you for bringing this up — we have captured these details in Section A.3 Details and Preprocessing on the AHMS Pre-training Dataset.
> - AHMS is an ongoing research study [1] sponsored by Apple and conducted in partnership with American Heart Association and the Brigham and Women’s Hospital, and we refer readers to the original paper on the AHMS [2] for further details on the dataset. Specifically Figure 8 in [2] has further information on general subject demographic distributions including age, body mass index, and self-reported race and ethnicity.  We have added a line to the paper specifying that this information is in this reference.
> - We use the raw 100hz accelerometry data as is, without specific preprocessing techniques, such as filtering, non-wear detection, or downsampling. Our choice of not filtering the data is supported by prior work in a daily monitoring setting [3], and unfiltered data can allow deep learning approaches to capture subtle nuances within the signal. Additionally, we aim to develop our methods to be robust to noise via augmentations, such as additive gaussian noise augmentations. We do not attempt to filter out periods of low accelerometry activity that may be indicative of non-wear detection. This is because small, minor changes in the accelerometer signal have been shown to still be informative, being able to predict heart rate [5].

---

> > ### Author Response · Authors · 2024-11-20
> > **Response to byiS (2/3)**
> >
> > **Table 3 and Overfitting Concerns (W3, W4, Q3)**
> >
> > - We have revised the paper to make the organization and claims from Table 3 more clear. The “MLP probe” results train a simple head on top of a frozen representation, and they show that RelCon FM provides a superior feature representation to Yuan et al.’s FM across the two down-stream tasks. In this case, there is clearly no overfitting.
> >   - Next, the “Fine-tuning” results initialize each model from the FM and train the model end-to-end in a fully-supervised fashion. These demonstrate the upper-bound of our FM for task-specific performance. In this way, both our RelCon FM and Yuan et al.’s FM undergo a task-specific overfitting to understand how well they can capture a task-specific representation.
> >   - Finally, the “From Scratch” results show the results after training the FM network backbones from a random initialization in a fully supervised fashion. The lower performance of these models in comparison to “Fine-tuning” is encouraging, as it tells us that our pre-trained representations do add value over simply relying on task-specific data.
> > - For further context, we utilize the same exact leave-one-subject-out cross validation splits Yuan et al. used for Opportunity and PAMAP2 by using their public dataset generation code [6,7], as well as utilizing their proposed MLP head with dimensionality of 512.  We verified our implementation with their public code release [8]. We have carefully checked our code again during this rebuttal process, and we cannot find any indication of any unexpected issues. These overfitting comments seem rooted in two ideas 1) abnormally high overall performance and 2) small performance increase from “From Scratch” to “Fine-tuned”:
> >   - 1) Abnormally High Overall Performance in Table 3:
> >     - Because these datasets utilized strict protocol requirements to ask users to conduct certain activities, the accelerometry data is generally quite clean, and it is not uncommon to achieve high >90 F1 scores, as seen below. Thus, we do not believe our results to be atypical.
> >       - [9] demonstrates 86.7 F1 on Opportunity and 96.44 F1 on PAMAP2
> >       - [10] demonstrates 98.3 F1 on PAMAP2
> >       - [11] demonstrates 96.04 F1 on Opportunity
> >   - 2) Small Performance Increase from “From Scratch” to “Fine-tuned” in Table 3:
> >     - The RelCon foundation model having a 5 difference in F1 score might perhaps make more sense in the context of our “From Scratch” scores, which were already quite high, at 94.0 F1 and 97.5 F1. The maximum F1 score is 100 and thus saturates at the upper end, with incremental increases being more difficult.
> >     - Yuan et al. is able to achieve a 10+ increase in F1 score since their “From Scratch” score started quite low at 38.3 F1 and 60.5 F1 for Opportunity and PAMAP2.
> >
> > **Generalizing to Time Lengths Beyond 2.56 seconds (Q2)**
> >
> > - From the literature, 2.56s is a common accelerometry subsequence size for motion tasks [12-16]. Additionally, we have shown that our approach can easily be used for time-series with other, differing lengths. In our comparisons against Yuan et al., 2024, we utilize a subsequence length of 10 seconds in order to match their evaluations, and we continue to show strong performance. In our comparisons against Haresamudram et al., 2022, we utilize a subsequence length of 2 seconds in order to match their evaluations. This highlights that our model is robust to varying input lengths and is generalizable across input data configurations, as would be desirable from a foundation model. This flexibility is enabled by the final temporal-global-average-pooling layer that we have at the end of our architecture. Additionally, in our AHMS workout-level classification task, we show how our method can be used with variable-length time-series that can last up to 10 minutes long by aggregating predictions across windows. We appreciate this valuable point which highlights a key functionality of foundation models, and we have added this discussion Section A.8 Generalizing to Time Lengths Beyond 2.56 seconds to re-emphasize this strength of our approach.

---

> > > ### Author Response · Authors · 2024-11-20
> > > **Response to byiS (3/3)**
> > >
> > > **Minor Paper Edits (Q1, Q7)**
> > >
> > > - Q1: “In text citations have not be been correctly phrased”
> > >   - Thank you for bringing this to our attention, and we have now fixed our in-text citations to consistently utilize \citep{} and \citet{} correctly, in accordance with the ICLR style guidelines.
> > > - Q7: “include a statement …”
> > >   - We have now added this to the captions of Table 3.
> > >
> > > **Bibliography**
> > >
> > > [1] <https://clinicaltrials.gov/study/NCT04198194?tab=table>
> > >
> > > [2] Shapiro, Ian, et al. "Pulse oximetry values from 33,080 participants in the Apple Heart & Movement Study." *NPJ Digital Medicine* 6.1 (2023): 134.
> > >
> > > [3] Campbell, Rhiannon A., et al. "Effects of digital filtering on peak acceleration and force measurements for artistic gymnastics skills." *Journal of Sports Sciences* 38.16 (2020): 1859-1868.
> > >
> > > [4] https://support.apple.com/guide/watch/lock-or-unlock-apple-watch-apd0e1e73b6f/watchos
> > >
> > > [5] Moebus, Max, et al. "Nightbeat: Heart Rate Estimation From a Wrist-Worn Accelerometer During Sleep." *arXiv preprint arXiv:2411.00731* (2024).
> > >
> > > [6] <https://github.com/OxWearables/ssl-wearables/blob/main/data_parsing/pamap.py>
> > >
> > > [7] <https://github.com/OxWearables/ssl-wearables/blob/main/data_parsing/oppo.py>
> > >
> > > [8] <https://github.com/OxWearables/ssl-wearables/blob/150550ea5d41800229c95e36f88f5bf0d2e7cf04/sslearning/models/accNet.py#L35>
> > >
> > > [9] Mekruksavanich, S., & Jitpattanakul, A. (2024). Device Position-Independent Human Activity Recognition with Wearable Sensors Using Deep Neural Networks. *Applied Sciences*, *14*(5), 2107.
> > >
> > > [10] Pang, Hongsen, Li Zheng, and Hongbin Fang. "Cross-Attention Enhanced Pyramid Multi-Scale Networks for Sensor-based Human Activity Recognition." *IEEE Journal of Biomedical and Health Informatics* (2024).
> > >
> > > [11] Xu, Hongji, et al. "Human activity recognition based on Gramian angular field and deep convolutional neural network." *IEEE Access* 8 (2020): 199393-199405.
> > >
> > > [12] Anguita, Davide, et al. "A public domain dataset for human activity recognition using smartphones." *Esann*. Vol. 3. 2013.
> > >
> > > [13] Chen, Yuqing, and Yang Xue. "A deep learning approach to human activity recognition based on single accelerometer." *2015 IEEE international conference on systems, man, and cybernetics*. IEEE, 2015.
> > >
> > > [14] Wang, Ning, et al. "Accelerometry based classification of walking patterns using time-frequency analysis." *2007 29th annual international conference of the ieee engineering in medicine and biology society*. IEEE, 2007.
> > >
> > > [15] McQuire, Jamie, et al. "Uneven and irregular surface condition prediction from human walking data using both centralized and decentralized machine learning approaches." *2021 IEEE International Conference on Bioinformatics and Biomedicine (BIBM)*. IEEE, 2021.
> > >
> > > [16] Mandong, Almontazer, and Usama Munir. "Smartphone based activity recognition using k-nearest neighbor algorithm." *Proceedings of the International Conference on Engineering Technologies, Konya, Turkey*. 2018.

---

> > > > ### Comment · Reviewer_byiS · 2024-11-22
> > > >
> > > > Thanks a lot of your thoughtful response and edits to the manuscript.
> > > >
> > > > Even though I still have concerns about the overfitting on your benchmarks, I guess that can be settled by other researchers upon the release of the model.
> > > >
> > > > Congrats on a nice paper. Happy to see another pre-trained model in the wearable community.
> > > > I have adjusted my score in light of your comments.

---

> > > > > ### Author Response · Authors · 2024-11-23
> > > > > **Thank you Reviewer byiS**
> > > > >
> > > > > Thank you for your kind words and support!

---

> > ### Comment · Reviewer_byiS · 2024-11-22
> >
> > Thank you this is clear.

---

### Official Review · Reviewer_afi1 · 2024-11-04

**Soundness:** 2
**Presentation:** 2
**Contribution:** 2
**Rating:** 5
**Confidence:** 4

**Summary:**

The current paper presents a foundation model for acceleration data on wrist wearables applied to human activity recognition. In this case, focus on the classification of a broad set of human activities (e.g, walking, stair-climbing, swimming,..).
The main contribution is an approach to include augmented data for enhancing the learning of similar samples systematically distorted. And a new formulation for a relative contrastive loss that encodes relative order relationships into the standard cross entropy loss.
The results of the foundational model proposed trained on 1 Billion segments of data of wrist acceleration (equivalent to 82 years of data from 87k participants).

**Strengths:**

1. A foundational model trained on a large data sample over 82 years of data from 87k participants
2. An interesting approach for training through self-supervised learning data augmentation.
3. Introducing a loss function (relative contrastive loss) that accounts for augmented points distance to the anchor for improving training, thus making it resistant to false positives.
4. Comparative analysis with similar approaches
5. Performance proved on the classification of motion tasks as well as regression of 2 metrics in walking.

**Weaknesses:**

1. The learning approach is only tested on only 2 datasets of a single sensor, thus the idea of an acceleration foundational model is not proven as a generalization is still questionable. Others such as: PAMAP2, USC-HAD, MHealth.
2. One suggestion would be to evaluate other acceleration-based HAR with multiple signals in benchmarks of HAR that go beyond single sensor-based.
3. Improvements to previous work seem marginal +0.01 AUC and 2.17 F1-score improvement for classification tasks on the AHMS dataset.
*3. It would be ideal to conduct statistical significance tests on these improvements and discuss the practical implications of these improvements. Also, please highlight the meaning of your improvement in some specific classes (does it improve greatly in one class?). Rather than relying only on F1 scores.
4. Results comparing to wrist acceleration models across the literature were limited to SimCLR and REBAR. And the analysis compared to supervised learning was minimally addressed.
*4 * I would recommend comparing the performance in some of the SOTA best-trained models for activity classification: e.g, F1-scores on PAMAP2 0.86 (Essa & Abdelmaksoud, 2023), MHealth: 0.94(Suh et al., 2023), 0.83 USC-HAD (Essa & Abdelmaksoud, 2023), and so.

References:
[1] Ehab Essa and Islam R. Abdelmaksoud. Temporal-channel convolution with self-attention network for human activity recognition using wearable sensors. Knowledge-Based Systems, 278:110867, 2023. ISSN 0950-7051. doi: https://doi.org/10.1016/j.knosys.2023.110867.
[2] Sungho Suh, Vitor Fortes Rey, and Paul Lukowicz. Tasked: Transformer-based adversarial learning for human activity recognition using wearable sensors via self-knowledge distillation. Knowledge- Based Systems, 260:110143, 2023. ISSN 0950-7051. doi: https://doi.org/10.1016/j.knosys. 2022.110143.

**Questions:**

1. What is your definition of a Motif,?
2. How could it be proven in this work that distance measures actually reflect similarities in motifs?
3. Why is the title "motion Foundation Model for wearable data" → your current work although extensive data was trained comprises "Wrist acceleration data" not any other modality even from IMU, thus it reads incorrectly.	(2,56 s segments of 100 Hz x,y,z accelerations)
4. In Figure 1 is there strict proof of all the characteristics listed here? I would like to see each of them analyzed more clearly in the conclusions.
5. What is the size of the dataset and classes used for evaluation on HHAR, and PAMAP2?
6. What is the distribution of data on each of these datasets? Balance on classes?
Is PAMAP2 wrist or ankle data used? There is a discrepancy between Table 3 and the description in the text.
7. What is the inference time, and computational complexity of the proposed approach RelCon? Is there any advantage over REBAR or SimCLR?
8.1 On Fig. 2, why is it concluded that the representation is clearer? I fail to see a difference compared with REBAR.
8.2. On the t-SNE, what perplexity was used? Are the results consistent at all levels?

---

> ### Author Response · Authors · 2024-11-20
> **Response to afi1 (1/3)**
>
> Thank you for your thoughtful and constructive feedback, and for recognizing the contribution of a large foundation model with contrastive loss evaluated on multiple downstream tasks.  We appreciate your insights, and are revising our manuscript and expanding our Appendix based on your feedback.
>
> **Further Clarification (W1, W3, W4).**  We appreciate you bringing these points to our attention, and apologize that they were not clearer in the original paper.
> - **W1**: “the learning approach is only tested on only 2 datasets”
>   - Thank you for raising this concern. To clarify, we tested on 6 distinct datasets (i.e. AHMS Classification Data, Gait Metrics Data, Opportunity, PAMAP2, HHAR, and Motionsense). We note that AHMS Classification Data and the Gait Metrics Data are two distinct datasets, collected with different protocols, participants, and settings.
>   - The number of datasets is now explicitly defined in the introduction, and we have also added additional language to the “Downstream Evaluation” section so that this is clear to readers.
> - **W3:** “marginal +0.01 AUC and 2.17 F1-score improvement for classification tasks on the AHMS dataset”
>   - We are revising the language of the paper to better emphasize that the strength of our results is demonstrating strong performance and generalizability across a variety of diverse tasks, not necessarily showing overwhelmingly strong improvements in each task. RelCon may have a closer margin with the second best model for a given task, but across tasks, the second best model consistently changes with RelCon consistently remaining the best. The specific example that you pointed out are the margins of RelCon compared to SimCLR for AHMS Subseq-level classification in Table 2. If we compare RelCon to SimCLR for AHMS Workout-level classification, RelCon has a much larger margin compared to SimCLR, and SimCLR is now the third best model.
>   - We feel that the consistency of our performance increases across the full breadth of evaluations of 6 datasets across 4 different tasks demonstrates the value of our approach, even though the gains are modest in a few cases.
> - **W4:** “Results comparing to wrist acceleration models across the literature were limited to SimCLR and REBAR”
>   - We compared our model to many other methodologies as well.  We have revised our text in Section 4.2 Downstream Evaluation to highlight that we compared our RelCon model against 11 models total: 3 pre-trained from scratch in “Task Diversity Evaluation” and 8 from the prior literature in “Benchmarking Evaluation”. Out of all of these models, there are 8 unique SSL approaches.
>     - In our Task Diversity Evaluation, we compare against three self-supervised models that we pre-trained from scratch on our dataset: Augmentation Prediction, SimCLR, and REBAR. This is seen in Tables 1, 2 and Figures 3, 4. Each of these methods are state-of-the-art SSL methods for accelerometry data.
>     - In our Benchmarking Evaluation, we compare against eight strategies: Augmentation Prediction, SimCLR, SimSiam, BYOL, MAE, CPC, and AE (Table 4) and against Yuan et al., 2024's foundation model (Table 3).
>   - We appreciate the point on including further SOTA supervised models, and we plan to look into including additional supervised models in our final camera ready. We would like to note that we focused on a single 3-axis accelerometer sensor in order to ensure broad generalizability. Please see the below section on “Using Accelerometer-Only Sensor Data” for further discussion.
>     - Thank you for suggesting the additional references. While these methods are not directly applicable to our work because they use multi-modal and multi-location streams of sensor data, we agree that these are important works to provide background context and have added them as references in Line 884.

---

> > ### Author Response · Authors · 2024-11-20
> > **Response to afi1 (2/3)**
> >
> > **Using Accelerometer-Only Sensor Data (W2, Q3)**
> > - Thank you for this discussion point, and it has been captured in Section A.4 Justification for Using Single-Sensor-Accelerometer-Only Data.
> > - It is true that motion information can be analyzed with multiple sensor streams, whether it be through multi-location (i.e. multiple accelerometer sensors strapped in different locations [1]) or multi-modal (i.e. extra IMU-based sensors, such as the gyroscopic sensor [2]) sensors. We recognize the value of a foundation model that can incorporate a multi-modality and/or multi-location stream of data, as it would enable for greater insights on human motion and physiology, and we are interested in pursuing this in future work.
> > - However, for our foundation model, we strive for broad generalizability in order to ensure that our learning approach and model is applicable across various settings. This includes low resource, single-modality settings that only have accelerometer sensors available. Gyroscopic sensors are quite power hungry [3], and accelerometer sensors are thus the most common and available sensor for monitoring human motion [4]. Additionally, we would like our model to also be applicable for real-world field settings, in which multi-location sensors are uncommon for daily usage due to their bulkiness and discomfort.
> > - As such, our foundation model is able to be benchmarked against a broad range of datasets (i.e. AHMS classification Data, Gait Metric Data, HHAR, Motionsense, PAMAP2, Opportunity), which each utilize different sensor setups, but all include at least one 3-axis accelerometer sensor.
> >
> > **Further Discussion on Our Method (W3, Q1, Q2, Q4, Q7)**
> > - **W3:** “please highlight the meaning of your improvement in some specific classes”
> >   - Thank you for the suggestion. We have amended Section 5.1’s Activity classification results to better explain this point. The ROC curves in Figure 4 and confusion matrix visualizations in Figure 5 show that RelCon is more accurate at classifying stair climbing compared to the other methods, which will often confuse stair climbing with climbing or elliptical. Each of these three classes has slower, deliberate hand swinging motions, and RelCon is better at distinguishing between them. We hypothesize that RelCon’s resistance to false positives and negatives enables the model to capture the subtle differences between them. Additionally, unlike REBAR and AugPred, RelCon can better predict outdoor running from indoor running. We hypothesize that RelCon can better identify when the running pattern is more uniform, implying that the running is perhaps being done indoors.
> > - **Q4, Q7**:  Comparison of RelCon against REBAR and SimCLR
> >   - This is a great discussion point. We have included an additional discussion based on your feedback in Section A.5 and have also updated our caption in Figure 1. False Negatives in a SimCLR-based model is a well-studied phenomena, with many recent works highlighting this issue and proposing novel methodologies to address this [6,7,8]. Accelerometry SimCLR models, as in [9] do not model within-user interactions. Instead they treat every subsequence as independent and hence do not capture within-user interactions. RelCon explicitly models within and between-user interactions by explicitly comparing an anchor against candidates from within-user across time and between-users. Unlike REBAR, RelCon seeks to model the relative positions of each of its candidates. Thus, RelCon is likely to be resistant to False positives and negatives due to a comparison that captures the nuanced differences between candidates.
> >
> > - **Q1, Q2**: Motifs
> >   - Here we use the term “motif” to refer to short temporal shapes within a sequence that may be indicative of semantic information. We utilize the motif framing to help explain how the dilated convolutions of our distance measure function can capture semantic information.
> >
> > - **Q7**: Computational Cost
> >   - During inference, each of our benchmarked methods has identical computation times because they each utilize ResNet-34 as the backbone architecture. The differences between them lie in the modified learning objective during training.
> >   - During training, RelCon needs to compare the relative distances of each candidate from the anchor with our distance function. The distance function utilizes a highly parallelizable transformer function with complexity of O(T^2 × d) [10] and a convolution to embed the inputs with a complexity of O(k × T x d^2) [10]. T=256 the subsequence length, d=64 the embedding dimension, and k=15 the kernel size. Therefore, because we calculate this distance function for every candidate, given a size of c, our total complexity during training is O(c × T^2 × d + c × k × T x d^2). In our future work, we aim to reduce the computational cost during training, and we have included this discussion point in our revised text in Section A.6 Computational Complexity of RelCon.

---

> > > ### Author Response · Authors · 2024-11-20
> > > **Response to afi1 (3/3)**
> > >
> > > **[CONTINUED] Further Discussion on Our Method (W3, Q1, Q2, Q4, Q7)**
> > > - **Q7**: “On Fig. 2, why is it concluded that the representation is clearer? I fail to see a difference compared with REBAR”
> > >   - Thank you for your question. We have clarified this in our text in the Figure 2 caption and in Section 5.1. RelCon’s t-SNE visualization shows a cluster of swimming labels on the bottom, to the right of the Indoor Cycling Cluster. REBAR, on the other hand, does not have this swimming cluster.  Additionally, REBAR’s Indoor Cycling Cluster is relatively less clustered together.
> > > - **Q7**: t-SNE perplexity
> > >   - Thank you for noting this, and we have clarified our Figure 2 caption to specify that the perplexity is set to 100 and remains constant across each visualized method. Trends in the visualization were consistent across perplexities.
> > >
> > > **Further Dataset Descriptions on “HHAR and PAMAP2” (Q5, Q6)**
> > >
> > > - Thank you for this clarifying question. In Section 4.2.2 Benchmarking Evaluation Datasets and Set-up, we have stated that our evaluation set-ups directly match the prior work of [5] and [11]. Based on this clarifying question, in our revised text, we have included full dataset descriptions in Section A.7.1 Evaluation Dataset Descriptions. We carefully ensured that our setup matched the prior works to enable these comparisons. We cross-referenced [5] with their public code release of their exact data split and generation [12]. Since [11] did not release their code publicly, we contacted the authors directly to ensure that we matched their exact splits and classes evaluated.
> > >   - For our comparison with [5], we utilize their setup: a 9-fold leave-one-subject-out cross validation for PAMAP2 (with the lying, sitting, standing, walking, ascending stairs, descending stairs, vacuum cleaning, and ironing classes).
> > >   - For our comparison with [11], we similarly follow exactly the same setup as in the original work. We a utilize 5-fold leave-subject-out cross validation for PAMAP2 (with the lying, sitting, standing, walking, running, cycling, nordic walking, ascending stairs, descending stairs, vacuum cleaning, ironing, and rope jumping classes) and HHAR (with the bike, sit, stairs down, stairs up, stand, and walk classes).
> > > - Because [5] uses the wrist data of PAMAP2, Table 3 reports the results evaluated on the PAMAP2 wrist data, and because [11] uses the leg data of PAMAP2, Table 4 reports the results evaluated on the PAMAP2 ankle data. Hence, we do not believe there is a discrepancy between Table 3 and the text description.
> > >
> > >
> > > Bibliography:
> > >
> > > [1] Jain, Yash, et al. "Collossl: Collaborative self-supervised learning for human activity recognition." *Proceedings of the ACM on Interactive, Mobile, Wearable and Ubiquitous Technologies* 6.1 (2022): 1-28.
> > >
> > > [2] García-de-Villa, Sara, et al. "Inertial sensors for human motion analysis: A comprehensive review." *IEEE Transactions on Instrumentation and Measurement* 72 (2023): 1-39.
> > >
> > > [3] <https://www.w3.org/TR/motion-sensors/>
> > >
> > > [4] Huang, Xinxin, et al. "Sensor-Based wearable systems for monitoring human motion and posture: A review." *Sensors* 23.22 (2023): 9047.
> > >
> > > [5] Yuan, Hang, et al. "Self-supervised learning for human activity recognition using 700,000 person-days of wearable data." *NPJ digital medicine* 7.1 (2024): 91.
> > >
> > > [6] Huynh, Tri, et al. "Boosting contrastive self-supervised learning with false negative cancellation." *Proceedings of the IEEE/CVF winter conference on applications of computer vision*. 2022.
> > >
> > > [7] Jin, Xiyuan, et al. "Time-Series Contrastive Learning against False Negatives and Class Imbalance." *arXiv preprint arXiv:2312.11939* (2023).
> > >
> > > [8] Chien, Jen-Tzung, and Kuan Chen. "False Negative Masking for Debiasing in Contrastive Learning." *2024 International Joint Conference on Neural Networks (IJCNN)*. IEEE, 2024.
> > >
> > > [9] Tang, Chi Ian, et al. "Exploring contrastive learning in human activity recognition for healthcare." *arXiv preprint arXiv:2011.11542* (2020).
> > >
> > > [10] Vaswani, A. "Attention is all you need." *Advances in Neural Information Processing Systems* (2017).
> > >
> > > [11] Haresamudram, Harish, Irfan Essa, and Thomas Plötz. "Assessing the state of self-supervised human activity recognition using wearables." *Proceedings of the ACM on Interactive, Mobile, Wearable and Ubiquitous Technologies* 6.3 (2022): 1-47.
> > >
> > > [12] https://github.com/OxWearables/ssl-wearables/blob/main/data\_parsing/pamap.py

---

> > > ### Author Response · Authors · 2024-11-25
> > > **Follow-up to Reviewer afi1**
> > >
> > > We would to follow up with you to check in to see if we have been able to address your concerns with our work. Please let us know if you would like any further clarification or discuss any other points. We appreciate your feedback and it has been already been used to greatly enhance the prior manuscript. Thank you very much!

---

> ### Author Response · Authors · 2024-11-30
> **Follow-up to Reviewer afi1 Before Discussion Period Ends**
>
> Dear Reviewer afi1: as the discussion period ends, we would like to follow-up one last time to check if we have been able to address your concerns. Thank you!

---

### Official Review · Reviewer_21L2 · 2024-11-04

**Soundness:** 4
**Presentation:** 3
**Contribution:** 3
**Rating:** 8
**Confidence:** 3

**Summary:**

This paper presents RelCon, a contrastive learning approach that leverages relative similarity between different accel signal segments (motifs) as a softened discriminative loss in contrastive learning. Authors use the proposed contrastive learning setup to pre-train a 1-D ResNet-34 backbone on 2.56 s long 100 Hz accel signal segments. The resulting pre-trained model generated a 256-dimensional embedding representing the accel signal segment. This pre-trained model was kept frozen and a linear layer was trained for downstream tasks, such as activity classification and gait metrics regression.

Authors show the effectiveness of their method via rigorous comparisons with SSL works (REBAR, AugPred, SimCLR) as well as supervised training approaches on the downstream classification tasks.

**Strengths:**

* The paper is very well-written. The authors clearly explain their approach, what parts are similar to the past work, and where they have improved over the past work.

* This paper has strong empirical results. I think, the approach is most similar to REBAR (which used motif-based similarity in contrastive loss) and AugPred (which pre-trained another accel-based foundation model). However, authors show that RelCon achieves much better empirical results compared to both REBAR and AugPred.

**Weaknesses:**

I think, the only weakness of the proposed method is that it can be seen as a somewhat straightforward extension of REBAR by i) using the relative distance as a soft metric instead of hard similarity labels, and ii) a few additional enhancements that were also proposed independently by others earlier (sec 3.1). However, authors clearly quantify the benefits of RelCon over earlier works via rigorous empirical evaluations where it outperforms AugPred, SimCLR, and REBAR in most cases.

**Questions:**

* Why is REBAR not included as a baseline in Table 4?

---

> ### Author Response · Authors · 2024-11-20
> **Response to 21L2 (1/1)**
>
> Thank you for the thoughtful and encouraging feedback. We appreciate your acknowledgement of the strength of the empirical results and the writing of our paper. We have also added a new experiment regarding the softness of our labels in the loss function and revised the paper according to your feedback.
>
> **W1**: RelCon as an extension
>
> - We appreciate your recognition that our extensive experiments show the strong empirical performance of RelCon.
> - However for i), we would like to note that using distance via our relative contrastive loss is particularly interesting because the relative contrastive loss is able to have a nice balance of softness. If we increase the softness in our loss function by utilizing a metric learning loss function [1], then this will hurt performance on the gait metric regression tasks. However, if we increase the hardness to a binary contrastive loss (i.e. REBAR), then this hurts performance across both gait metric regression and activity classification. Please see the table below, and we have added this discussion to Section A.9.
> ||Velocity|DST|AHMS-Subseq|AHMS-Workout|
> | :- | :-: | :-: | :-: | :-: |
> ||↑ Corr|↑ Corr|↑ F1|↑ F1|
> |RelCon|0\.8431|0\.7559|38\.56|55\.28|
> |w/ Softer Metric Loss [1]|-3.64%|-6.11%|-1.45%|1\.39%|
> |w/ Harder Binary Contrastive Loss|-6.86%|-9.82%|-5.63%|-9.03%|
>
>
> - Additionally for ii), the enhancements are interesting as they help demonstrate novel ways of designing a distance function that can learn to capture abstract, semantic information (i.e. 3D rotation invariance).
>
> **Q1**: Why is REBAR not included as a baseline in Table 4?
>
> - In order to form a fair comparison against REBAR, we pre-train REBAR and RelCon under the same exact conditions (e.g. pre-train dataset, training steps, backbone architecture, etc.) and compare them in Tables 2 and 3.
> - In Table 4, we compare our RelCon FM against the FMs presented in Haresamudram et al., 2022, that used completely different pre-training conditions. This is done in order to show how our RelCon FM model compares against prior FMs that were evaluated on public datasets. REBAR was not used as a baseline in the original Haresamudram et al., 2022 study. We have revised Table 4 to better reflect that the baselines originate from a reference to prior work.
>
> Bibliography:
>
> [1] Kim, Sungyeon, et al. "Deep metric learning beyond binary supervision." *Proceedings of the IEEE/CVF Conference on Computer Vision and Pattern Recognition*. 2019.

---

### Author Response · Authors · 2024-11-20
**Response to All Reviewers**

We would like to thank the reviewers for their insightful feedback on our work. We would also like to share that we will be releasing our code publicly upon publication, which will allow readers to easily re-train our FM training approach in addition to providing a unified benchmarking task to guide model development. In response to the feedback and suggestions, we have revised our paper accordingly (shown as blue text in the paper) and have included two new experiments, one demonstrating how increasing or decreasing the softness of the labels in the RelCon loss function affects performance and another that demonstrates how our RelCon FM can continue to outperform another large-scale pre-trained FM from Yuan et al., 2024 when controlling for the backbone encoder’s architecture.


We sincerely appreciate that reviewers recognized key contributions of our work: the novel methodology and the significance of our empirical results. We are glad that reviewers recognized the novelty of our approach that combines an augmentation-based motif-similarity with a relative contrastive loss, and the value of our findings using a model trained at scale across a variety of tasks. Please see specific responses to each individual reviewer below, we are happy to continue the discussion and address any further questions.

---

### Meta-Review · Area_Chair_exEq · 2024-12-12

**Metareview:**

RelCon is a self-supervised learning method designed for motion foundation models using wearable sensor data. It employs a learnable distance measure combined with a softened contrastive loss to capture motif similarities and domain-specific semantics, including rotation invariance. Trained on a large dataset of 1 billion accelerometer time-series segments from 87,376 participants, the model demonstrates strong generalizability across various downstream tasks such as human activity classification and gait metric regression. Without requiring fine-tuning, RelCon surpasses state-of-the-art methods in both classification and regression, establishing a new standard for foundation models in wearable motion data analysis.

The paper has a novel approach and the empirical results demonstrate its effectiveness strongly in 6 datasets. The only concern now is the reproducibility of this paper. The authors should open-source according to their claims. No major flaws are found before and after the rebuttal. I believe the paper should be accepted.

**Additional Comments On Reviewer Discussion:**

Though reviewers have positive attitudes of the paper, they still have many concerns:
1. **Limited Generalization and Benchmarking**: The evaluation is confined to two datasets from a single sensor type, raising doubts about the generalizability of the foundational model. Broader benchmarking on diverse datasets like PAMAP2, USC-HAD, and MHealth with multiple signals is necessary. Additionally, comparisons are limited to specific models like SimCLR and REBAR, while other state-of-the-art methods for activity classification remain unexplored.

2. **Marginal Performance Improvements and Lack of Statistical Significance**: Improvements over prior methods, such as a +0.01 AUC and 2.17 F1-score for classification tasks on AHMS, are minimal. Statistical significance tests and discussions on the practical implications of these gains, along with class-specific performance breakdowns, are missing.

3. **Concerns Over Data and Methodology**: The use of consumer-grade devices in the Apple Heart and Movement Study introduces noise and missing data, warranting more descriptive analyses of the dataset and preprocessing steps. Additionally, results suggest possible overfitting, especially when compared to benchmarks on smaller datasets like Opportunity and PAMAP2.

4. **Reproducibility and Fairness in Comparisons**: Neither the code nor the model is open-source, hindering reproducibility. Comparisons with prior work are not entirely fair, as differences in model architectures (ResNet-34 vs. ResNet-18) introduce variability. A fair evaluation should standardize architectures to isolate the impact of the proposed SSL method. Furthermore, more comprehensive benchmarking against established datasets for SSL in accelerometer data, such as Capture-24, is needed.

Apart from the pending opensource, the authors provide new results for (1)(2) and discussions with sufficient references for (3). I believe most concerns are addressed. Since the reviewer afi1 (the only below borderline reviewer) does not provide response though I send many reminders and emails, I will weigh this reviewer lower. The average score indicates that it is a clear accept.

---

### Decision · Program_Chairs · 2025-01-22

Accept (Poster)